# Exp-$\alpha$: Beyond Proportional Aggregation in Federated Learning

## Abstract

Federated Learning (FL) is a distributed learning paradigm, which computes gradients of a model locally on different clients and aggregates the updates to construct a new model collectively. Typically, the updates from local clients are aggregated with weights proportional to the size of clients' local datasets. In practice, clients have different local datasets suffering from data heterogeneity, such as imbalance. Although proportional aggregation still theoretically converges to the global optimum, it is provably slower when non-IID data is present (under convexity assumptions), the effect of which is exacerbated in practice. We posit that this analysis ignores convergence *rate*, which is especially important under such settings in the more realistic non-convex real world. To account for this, we analyze a generic and *time-varying* aggregation strategy to reveal a surprising *trade-off* between convergence rate and convergence error under convexity assumptions. Inspired by the theory, we propose a new aggregation strategy, Exp-$\alpha$, which weights clients differently based on their severity of data heterogeneity. It achieves stronger convergence rates at the theoretical cost of a non-vanishing convergence error. Through a series of controlled experiments, we empirically demonstrate the superior convergence behavior (both in terms of rate and, in practice, even error) of the proposed aggregation on three types of data heterogeneity: imbalance, label-flipping, and domain shift when combined with existing FL algorithms. For example, on our imbalance benchmark, Exp-$\alpha$, combined with FedAvg, achieves a relative $12\%$ increase in convergence rate and a relative $3\%$ reduction in error across four FL communication settings.

## 1 Introduction

Federated Learning (FL) (McMahan et al., 2017) is a decentralized approach for learning a model on distributed data to preserve data privacy. Because data reside on clients and are never transmitted to a central server, privacy is preserved. However, data on local clients are often correlated with their demographics and preferences. This makes training data highly non-IID or heterogeneous (Wang et al., 2021; Zhang et al., 2021; Kairouz et al., 2021), containing label imbalance, noisy labels (e.g. label-flipping), or domain shift. This can significantly impact a model's performance and specifically convergence rates (Zhao et al., 2018; Li et al., 2019). To tackle the issue of data heterogeneity, the majority of federated learning have focused on improving the local optimization (Zhao et al., 2018; Shoham et al., 2019; Karimireddy et al., 2020; Zhang et al., 2020; Acar et al., 2021) and the global optimization (Hsu et al., 2019; Reddi et al., 2020) objectives in a federated learning pipeline. Few papers have paid attention to the other aspects of federated learning, such as client selection (Cho et al., 2020) and model aggregation (Chen et al., 2020; Wang et al., 2020).

Most existing methods use proportional aggregation (McMahan et al., 2017), whose aggregation weights are proportional to the size of local dataset. Although proportional aggregation still theoretically converges when non-IID data is present under convexity assumptions, we posit that this analysis ignores convergence *rate*, which is especially important under such settings in the real world, because proportional aggregation assumes equal importance of all samples. Intuitively, non-IID data makes the equal importance property questionable since imbalanced data can bias predictions towards majority classes, and noise or domain shift can slow down convergence.

To study this, we start by introducing the proportional aggregation strategy and discussing its merits: *equal importance* and *asymptotic convergence*. Following prior works (Wang et al., 2021; Reisizadeh et al., 2020; Yuan et al., 2021), we define the federated global objective $F(\mathbf{W})$ of the server as a weighted sum of $N$ local objectives $F_i(\mathbf{W})$ in Eq. 1. $F(\cdot)$ denotes generic loss/risk function.

**Definition 1**

$$F(\mathbf{W}) := \sum_{i=1}^{N} \rho_i F_i(\mathbf{W}) \quad where \quad F_i(\mathbf{W}) = \mathbb{E}_{\xi \sim P_i}\left[f(\mathbf{W}; \xi)\right]. \tag{1}$$

$\sum_{i=1}^{N} \rho_i = 1$ are the aggregation weights, $\mathbf{W} \in \mathbb{R}^d$ denotes the global model and $\xi$ is a sample from the local data distribution $P_i$. Usually in the distributed learning (Stich, 2018) and federated learning (Li et al., 2019) literature, the weights are set to be proportional to the number of samples on a client denoted as $\rho_i = \frac{|\Xi_i|}{\sum_{j=1}^{N} |\Xi_j|}$, where $|\Xi_i|$[1] is the size of local dataset $\Xi_i$. This weighting scheme has an intuitive interpretation, i.e., the global data can be equivalently seen as the union of local datasets, and the federated global objective is equivalent in expectation to what one would optimize centrally if data are sampled randomly from it. Proportional aggregation is then used to compute an *unbiased* update to Eq. 1. In summary, proportional aggregation is a statistically sound strategy, giving all data points *equal* importance and providing *asymptotic* convergence to a hypothetical centralized objective, i.e., achieving zero-error eventually. However, a recent survey calls these properties into question (Wang et al., 2021). In the real world, the defining characteristics of proportional aggregation, particularly equal importance and asymptotic convergence, can be less well justified. The property of equal importance of all participating data can be less desirable when data heterogeneity is severe. For example, even though the convergence of using proportional aggregatoin (with zero-error under convex settings) with non-IID clients is guaranteed, it is provably slower (Li et al., 2019), and with data poisoning (such as label-flipping), it can be even unstable (Xie et al., 2019; Jebreel et al., 2022). This is exacerbated by the limited communication rounds in FL, making the asymptotic convergence property less relevant since asymptotic convergence can only be achieved under the assumption of unlimited communication. As a result, two algorithms with comparable asymptotic convergence can perform quite differently in practice (Wang et al., 2021).

In this paper, we study a generic and *time-varying* aggregation strategy, $\sum_{i=1}^{N} \rho_i^t = 1$, where $\rho_i^t$ is the weight for client $i$ at time $t$, as opposed to proportional aggregation. A theoretical study of this strategy reveals a surprising *trade-off* between convergence rate and convergence error, allowing us to make more explicit what proportional aggregation favors and to develop new algorithms that make different trade-offs. For example, proportional aggregation, when instantiated in our framework as a special case, is shown to favor convergence error at the cost of convergence rate. More specifically, we start from a theoretical analysis on the convergence of FedAvg (McMahan et al., 2017), a prototypical FL algorithm, while allowing the aggregation weights to change over time. The resultant convergence bound in this more generic setting reveals a family of aggregation strategies that 1) improves convergence rate but 2) leaves a theoretically non-vanishing error w.r.t the proportionally weighted federated objective (Eq. 1). Subsequently, we propose a specific aggregation strategy in this family, Exp-$\alpha$, which weights clients differently based on their severity of data heterogeneity and can achieve stronger convergence rates at the theoretical cost of a non-vanishing convergence error. Intuitively, this strategy puts larger weights on clients sharing more similar data distribution to each other. Empirically, we go beyond theory to test its effectiveness on three major types of local data heterogeneity: *imbalance* (Zhao et al., 2018), *label-flipping* (Xie et al., 2019) and *domain shift* (Li et al., 2021). Our results suggest that an aggregation strategy with faster convergence rate can be more important in practice than one with theoretically zero-error under the convex assumption; in practice, our method achieves both better rates *and* better errors, owing to the fact that practical settings are non-convex. For example, on our imbalance benchmark, Exp-$\alpha$, combined with FedAvg, achieves a relative $12\%$ increase in convergence rate and a relative $3\%$ reduction in error across four FL communication settings. To sum up, our contributions are:

- We analyze the convergence of FedAvg with a generic and time-varying aggregation strategy to reveal a trade-off between convergence rate and error under convexity assumptions, and elucidate properties of prior proportional aggregation strategies.

---

[1] We use the notation $|\cdot|$ to denote the size of a set.

- We propose a new aggregation strategy, Exp-$\alpha$, that trades convergence rate over error under convexity assumptions. When applied to several existing FL algorithms in real world experiments, Exp-$\alpha$ demonstrates superior performance in both convergence rate and error over the widely used proportional aggregation on three types of data heterogeneity.

## 2  BACKGROUND AND RELATED WORKS

**Federated learning (FL)** is a distributed machine learning paradigm developed to preserve privacy while enabling continual development of an ML model on private data (McMahan et al., 2017). FL generally consists of three stages: client selection, client update and server update. Most FL algorithms innovate on one component of this algorithm.

**Client Selection**: At the beginning of a round of communication (a global iteration), the current global model $\mathbf{W}^t \in \mathbb{R}^d$ is distributed to a randomly sampled set of $N$ local clients from a large pool of candidates $\mathcal{N}$, sampled from a population distribution $\mathcal{C}$ supported on $\mathcal{N}$. If $N < |\mathcal{N}|$, then this is called partial-participation. Most paper follows a uniform client sampling strategy (Li et al., 2019). However, a recent work (Cho et al., 2020) shows that a biased strategy can bring pratical improvement to FL algorithms.

**Client Update**: After receiving the global model, the clients optimize their copies of it *independently* on their own local data $\xi \in \Xi_i \sim P_i(X, Y)$, where $\xi$ represents an element in the set of local data $\Xi_i$ on client $i$ sampled uniformly from the local data distribution $P_i$, for a specified $E$ number of steps to arrive at *different* updated local models $\mathbf{W}_i^{t+E} \in \mathbb{R}^d$ for $i \in \{1, ..., N\}$. This is the most investigated stage in FL research due to its unique non-IID (distribution shift) challenge (Zhao et al., 2018; Li et al., 2019). The vanilla FedAvg (McMahan et al., 2017) uses plain SGD updates, which can only handle mild non-IID data. Many followup works design regularization techniques to improve convergence under more severe distribution shift. Please see Appendix A.1 for an introduction to those methods. Our contribution is *orthogonal* to FL research in this category and can be combined. We will demonstrate this compatibility in our experiments (Sec. 4).

**Server Update:** To complete this round of communication, the updated local models are sent back to the central server for aggregation, which yields the next global model. Server update can be split into two steps: *aggregation* and *optimization* (Reddi et al., 2020). Our work focuses on the aggregation step in the server update stage. Specifically, aggregation refers to how gradients are combined and optimization refers to how the aggregated gradients are applied. Please see Appendix A.1 for an introduction to FL algorithms with different server optimization techniques. Few have studied the aggregation step in server update. FOCUS (Chen et al., 2020) measures the performance of a local model on a globally shared dataset and assigns an aggregation weight accordingly. However, the requirement of a global dataset that encompasses unknown local data distributions violates the privacy premise of FL. A recent work (Wang et al., 2020) discovered an implicit bias in aggregation when the number of local updates is different and proposed a mitigation strategy. This problem is orthogonal to our target on non-IID data and therefore, we keep the number of local updates the same on all clients in our experiments. Nonetheless, most existing works use proportional aggregation. Our contribution is *orthogonal* and *compatible* to other innovations in the optimization step. We will demonstrate this compatibility in the experiment section (Sec. 4).

## 3  GOING BEYOND PROPORTIONAL AGGREGATION

Existing FL convergence analyses often assume proportional aggregation in their deviation (Li et al., 2019; Khaled et al., 2020). This strategy yields asymptotically zero-error convergence under convex settings. However, as we will show in this section, revisiting the convergence analysis with a generic, time-varying aggregation strategy reveals that by carefully designing the aggregation weights, one can theoretically trade off convergence rate over error. This section is organized as the following. Sec. 3.1 introduces several common assumptions in FL convergence analysis and notations necessary to understand the theoretical results; Sec. 3.2 presents a convergence bound with a generic and time-varying aggregation strategy; Sec. 3.3 discusses the trade-off between convergence rate and error with a derived corollary; Finally, inspired by the corollary, Sec. 3.4 proposes a practical aggregation strategy, Exp-$\alpha$, which demonstrates superior convergence behavior in terms of both rate and error on several benchmarks in Sec. 4.

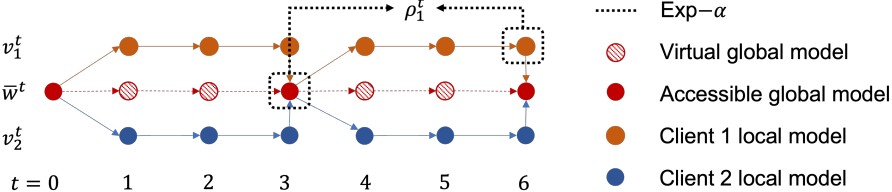

Figure 1: Illustration of our proposed Exp-$\alpha$ (Sec. 3.4) with three local optimization steps, i.e., $E = 3$, and two clients. In this example, synchronization/communication steps are $t = 0, 3, 6$. Exp-$\alpha$ calculates the aggregation weights based on the latest accessible global model and local models.

### 3.1 Time-Varying Aggregation: Assumptions and Notations

We first make some common assumptions in FL convergence analysis.

**Assumption 1** *The local objective functions $F_1, ...F_N$ are $\mu$-strongly convex: $F_i(\mathbf{W}) - F_i(\mathbf{V}) \geq (\mathbf{W} - \mathbf{V})^T \nabla F_i(\mathbf{V}) + \frac{\mu}{2}\|\mathbf{W} - \mathbf{V}\|_2^2 \quad \forall \mathbf{W}, \mathbf{V}$.*

**Assumption 2** *The local objective functions, $F_1, ..., F_N$ are $\mathcal{L}$-smooth functions: $F_i(\mathbf{W}) - F_i(\mathbf{V}) \leq (\mathbf{W} - \mathbf{V})^T \nabla F_i(\mathbf{V}) + \frac{L}{2}\|\mathbf{W} - \mathbf{V}\|_2^2 \quad \forall \mathbf{W}, \mathbf{V}$.*

**Assumption 3** *Bounded local gradient variance, let $\xi_i \sim P_i$ be a sampled data point on client $i$. The variance of gradients on all devices is bounded: $\mathbb{E}\|\nabla f_i(\mathbf{W}^t; \xi_i) - \nabla F_i(\mathbf{W}^t)\|^2 \leq \sigma_i^2 \quad \forall i \in \{1, ..., N\}$.*

**Assumption 4** *Bounded local gradients, let $\xi_i \sim P_i$ be a sampled data point on client $i$. The squared norm of gradients on all devices are bounded: $\mathbb{E}\|\nabla f_i(\mathbf{W}^t; \xi_i)\|^2 \leq G^2 \quad \forall i \in \{1, ..., N\}$.*

In addition to the convex Assumption 2, Assumptions 1-4 are fairly common assumptions in non-convex optimization literature (Reddi et al., 2016; Ward et al., 2020) and federated learning literature (Li et al., 2019; Cho et al., 2020). There are other FL works relaxing the above assumptions. For example, FedAdaGrad (Reddi et al., 2020) relaxes the convex assumptions and shows that the expected gradient goes to zero, thus converging to a stationary point with unknown error bound. While it's sufficient to demonstrate the hidden convergence dependency on aggregation weights under the current assumptions, extending our subsequent analysis to different FL assumptions can be an interesting future work.

We will utilize the method of virtual sequence (Stich, 2018) for the proof of the main theorem. Let $\mathcal{I}_E$ be the set of synchronization/communication steps, such that $\mathcal{I}_E = \{n \times E | n = 0, ..2\}$, where $E$ denotes the number of local update iterations. The virtual sequence $\bar{\mathbf{W}}^{t+1}$ is defined as:

$$\bar{\mathbf{W}}^{t+1} = \sum_{i=1}^{N} \rho_i^{t+1} \mathbf{W}_i^{t+1} \quad \text{where} \quad \mathbf{W}_i^{t+1} = \left\{ \begin{array}{ll} \mathbf{V}_i^{t+1}, & \text{if } t + 1 \notin \mathcal{I}_E \\ \sum_{i=1}^{N} \rho_i^{t+1} \mathbf{V}_i^{t+1}, & \text{if } t + 1 \in \mathcal{I}_E \end{array} \right\}. \quad (2)$$

where $\rho_i^{t+1} \geq 0$ and $\sum_{i=1}^{N} \rho_i^{t+1} = 1$ is the time-varying aggregation weight, and $\bar{\mathbf{W}}^0 = \mathbf{W}_i^0 = \mathbf{W}^0$. $\mathbf{V}_i^t$ denotes the local model $i$ at optimization step $t$. In reality, we only have access to $\bar{\mathbf{W}}^{t+1}$ when $t + 1 \in \mathcal{I}_E$, i.e., time of actual synchronization. When this happens, we write $\mathbf{W}^T$ where $T \in \mathcal{I}_E$. We provide a graphical illustration of the virtual sequence in Fig. 1, which also features our proposed method, Exp-$\alpha$. Furthermore, let $\mathbf{W}^*$ be the optimal solution to the federated global objective in Eq. 1, i.e., $\mathbf{W}^* = \arg\min_{\mathbf{W}} F(\mathbf{W}) = \arg\min_{\mathbf{W}} \sum_{i=1}^{N} \rho_i F_i(\mathbf{W})$, and $\mathbf{W}_i^*$ be the optimal solution to a client's data distribution, i.e., $\mathbf{W}_i^* = \arg\min_{\mathbf{W}} F_i(\mathbf{W})$.

### 3.2 Time-Varying Aggregation: Conservative Error Bound

Taking into account of generic and time-varying aggregation weights, $\rho_i^t$, we present the following theorem for the FedAvg algorithm (McMahan et al., 2017), a prototypical FL algorithm that uses vanilla SGD for local and global updates and proportional aggregation in the original work.

**Theorem 1** *Assume Assumptions [1]- [4] hold and $L, \mu, \sigma_i, G$ be defined therein. Choose $\gamma = \max \frac{L}{\mu}$ and the learning rate $\eta_t = \frac{2}{\mu(\gamma+t)}$ and $T \in \mathcal{I}_E$. Then FedAvg using SGD with full device participation and a generic, time-varying aggregation weights $\sum_{i=1}^{N} \rho_i^t = 1$ satisfies*

$$\mathbb{E}[F(\mathbf{W}^T)] - F(\mathbf{W}^*) \leq \underbrace{\frac{L}{(\gamma+T)}\left(\frac{2\bar{B}}{\mu^2} + \frac{\gamma}{2}\Delta_0\right)}_{vanishing} + \underbrace{\frac{L}{\mu}(\Gamma - \Omega)}_{non-vanishing}$$

*where* $\Delta_0 = \|\mathbf{W}^0 - \mathbf{W}^*\|_2^2$, $\bar{B} = \max_t(\sum_{i=1}^{N}(\rho_i^{t+1})^2 \sigma_i^2) + 8(E-1)G^2 + 6L\Omega$, $\Gamma = \max_t \sum_{i=1}^{N} \rho_i^{t+1}(F_i(\mathbf{W}^*) - F_i(\mathbf{W}_i^*))$, *and* $\Omega = \min_t \sum_{i=1}^{N} \rho_i^{t+1}(F_i(\bar{\mathbf{W}}^t) - F_i(\mathbf{W}_i^*)) \forall t \geq 0$.

We provide a complete proof of the main theorem in [A.9]. The convergence bound in Thm. [1] has two outstanding components, a vanishing term decreasing over time and a non-vanishing term. The vanishing term goes to zero with time and controls convergence rate; the non-vanishing term does not decrease over time and results in a non-zero error after convergence. The convergence result is consistent with convergence bound using proportional aggregation (Li et al., 2019). Specifically, we can show (Appendix [A.13]) that if $\rho_i^t = \rho_i = \frac{|\Xi_i|}{\sum_{j=1}^{N}|\Xi_j|}$, then $\Omega = \Gamma = \sum_{i=1}^{N} \rho_i(F_i(\mathbf{W}^*) - F_i(\mathbf{W}_i^*))$, and the non-vanishing error is zero. This demonstrate that proportional aggregation, as a special case of our more general analysis, favors convergence error at the cost of convergence rate.

## 3.3 TIME-VARYING AGGREGATION: TRADE-OFF BETWEEN SPEED AND ERROR

From Thm. [1], we observe that FedAvg with a generic time-varying weighing converges at a rate of

$$\mathcal{O}\left(\frac{\Omega}{T}\right) + \mathcal{O}(\Gamma - \Omega). \quad \text{where} \quad \Omega = \min_t \sum_{i=1}^{N} \rho_i^{t+1}(F_i(\bar{\mathbf{W}}^t) - F_i(\mathbf{W}_i^*))$$

The quantity $\Omega$ controls the convergence rate and the non-vanishing error. The intuitive way to improve the convergence rate is to design a strategy which gives a *small* $\Omega$. Let $\Omega_{pr}$ denote the quantity $\Omega$ defined by proportional aggregation, i.e., $\Omega_{pr} \triangleq \min_t \sum_{i=1}^{N} \rho_i(F_i(\bar{\mathbf{W}}^t) - F_i(\mathbf{W}_i^*))$, where $\rho_i = \frac{|\Xi_i|}{\sum_{j=1}^{N}|\Xi_j|}$. Specifically, if the goal is to improve over proportional aggregation, then the proposed aggregation strategy should lead to an $\Omega$ *smaller* than $\Omega_{pr}$. Upon close examination of $\Omega$ in Thm. [1], one can expect that the key lies in choosing the aggregation weight $\rho_i^{t+1}$ according to the relative magnitude of the quantity $\Omega_i^t \triangleq F_i(\bar{\mathbf{W}}^t) - F_i(\mathbf{W}_i^*)$ for all $i \in \{1, ..., N\}$. To this end, we provide the following corollary to formally justify the intuition. We show that with a specific choice of $\rho_i^{t+1}$, one can achieve $\Omega \leq \Omega_{pr}$.

**Corollary 1.1** *Assume* $\frac{N|\Xi_i|}{\sum_{j=1}^{N}|\Xi_j|}(F_i(\bar{\mathbf{W}}^t) - F_i(\mathbf{W}_i^*))$ *are arranged in decreasing order, i.e.,* $\frac{N|\Xi_i|}{\sum_{j=1}^{N}|\Xi_j|}(F_i(\bar{\mathbf{W}}^t) - F_i(\mathbf{W}_i^*)) \geq \frac{N|\Xi_{i+1}|}{\sum_{j=1}^{N}|\Xi_j|}(F_{i+1}(\bar{\mathbf{W}}^t) - F_{i+1}(\mathbf{W}_{i+1}^*))$. *If we choose*

$$\rho_i^{t+1} \propto \frac{N|\Xi_i|}{\sum_{j=1}^{N}|\Xi_j|}\mathcal{U}\left(\frac{N|\Xi_i|}{\sum_{j=1}^{N}|\Xi_j|}\left(F_i(\mathbf{W}_i^*) - F_i(\bar{\mathbf{W}}^t)\right)\right) \quad \forall t,$$

*where* $\mathcal{U}(*) \geq 0$ *is a non-decreasing function, then* $\Omega \leq \Omega_{pr}$.

A detailed proof is provided in Appendix [A.11]. In Corollary [1.1], $\bar{\mathbf{W}}^t$ is the virtual global model (Eq. [2]) at the previous step and $\mathbf{W}_i^*$ is the optimal local model for client $i$. Therefore, the quantity $\Omega_i^t \triangleq F_i(\bar{\mathbf{W}}^t) - F_i(\mathbf{W}_i^*)$ captures the performance difference between the *closest* virtual global model and the *optimal* local model on client $i$. Intuitively, one would expect that a client with more severe distribution shift will result in a larger $\Omega_i^t$ since the current virtual global model should not work well on this severely shifted distribution, resulting in a larger discrepancy between $F_i(\bar{\mathbf{W}}^t)$ and $F_i(\mathbf{W}_i^*)$. *In other words, the aggregation strategy in Corollary [1.1] puts smaller weights on more severely shifted clients based on current performance difference.* Consequently, it is reasonable to expect a trade off between convergence speed and convergence error depending on how aggressively the algorithm down-weights shifted clients. In our experiments, to avoid the compound issue of update bias due to unequal number of local updates (Wang et al., 2020), we deliberately keep the size of local datasets equal, i.e., $|\Xi_i| = |\Xi_j|$. Therefore $\frac{N|\Xi_i|}{\sum_{j=1}^{N}|\Xi_j|} = 1$.

### 3.4 TIME-VARYING AGGREGATION: THE EXPONENTIAL FUNCTIONS

One family of functions that satisfies $\mathcal{U}(*) \geq 0$ and non-decreasing, is the exponential functions. We will use this as the paramterization for our empirical investigation. However, at a synchronization step $T \in \mathcal{I}_E$, according to Thm. 1 and Corollary 1.1, we need to evaluate $F_i(\bar{\mathbf{W}}^{T-1})$ and $F_i(\mathbf{W}^*)$ to calculate the aggregation weights. This is not realistic, however; first, we only have access to the virtual global model at $t = T - E$, i.e., the model from the previous synchronization step since the current one has yet to be calculated. Therefore, the *closest* available global model is $\bar{\mathbf{W}}^{T-E}$. Second, in the most common setting, FL algorithms only optimize a local model for fixed number of $E$ epochs and do not train it to convergence[2]. Thus, the *closest* approximation to $F_i(\mathbf{W}^*)$ is the current local model after $E$ local updates from the previous synchronization, $F_i(\mathbf{W}_i^T)$. A graphical illustration is provided in Fig. 1. For subsequent investigation, we use the following approximation,

$$\rho_i^T \propto \exp\left( \frac{F_i(\mathbf{W}_i^T) - F_i(\bar{\mathbf{W}}^{T-E})}{\alpha} \right), \tag{3}$$

where $\alpha$ is a temperature hyperparameter to control the strength of the proportionality. As $\alpha \to \infty$, the strength of proportionality decreases, e.g., in the limit, $\rho_t \to 1$. Intuitively, a small $\alpha$ increases the concentration of $\rho_i^t$ and a large $\alpha$ decreases the concentration and makes the weights more evenly spread. In subsequent sections, we term this family of aggregation strategy as the Exp-$\alpha$ method. We provide a detailed algorithm description of Exp-$\alpha$ and discussion on computation in Appendix A.2.

## 4 EXPERIMENTS

**Overview.** In this section, we present experiments to test the capability of the Exp-$\alpha$ strategy beyond theory. To this end, we surveyed existing literature and identified three dominant data heterogeneity types: *imbalance* (Zhao et al., 2018), *label-flipping* (Xie et al., 2019) and *domain shift* (Li et al., 2021). Each types of heterogeneity brings a specific challenge to an aggregation strategy. Specifically, imbalanced clients require the aggregation to be adaptive to the severity of imbalance; label-flipping requires the aggregation to block contributions from label-flipped clients; domain shift requires the aggregation to *not* disregard any clients since all domains should contribute.

**Datasets.** To benchmark on different data heterogeneity, we use popular datasets in FL research (Zhao et al., 2018; Li et al., 2021; Yuan et al., 2021). For imbalance, we adopt the popular Imbalanced CIFAR10 (Cao et al., 2019) setting in the imbalanced classification task. For label-flipping experiments, we use CIFAR10 with randomly flipped labels (Xie et al., 2019; Jebreel et al., 2022). For domain shift experiments, following Li et al. (2021), we use Digits (Li et al., 2021), Office-Clatech10 (Gong et al., 2012), and DomainNet (Peng et al., 2019), each of which consists of a range of different domains with shared labels. Specifically, Digits has five domains, Office-Clatech10 has four domains, and DomainNet has six domains. Please see Appendix A.4 for details.

**Metrics.** We compare different methods using the accuracy achieved at both the halfway and full global training iterations (McMahan et al., 2017). Higher accuracy means lower converged error using the same number of optimization steps. We also report the global iterations required to achieve $X$ performance (given as "Acc$x$") (McMahan et al., 2017). Lower "Acc$x$" means that the algorithm converges to the same performance using fewer rounds of global communications and has higher convergence rate. We separate datasets into train, validation, and test splits, and report both validation and test accuracy if applicable in our experiments.

**Backbone Algorithms.** Exp-$\alpha$ is an aggregation strategy and can plug into most existing FL methods. We select five representative baselines from different categories as the backbone algorithm: FedAvg (McMahan et al., 2017), FedAvgM (Hsu et al., 2019), FedAdam (Reddi et al., 2020), Fed-Prox (Zhao et al., 2018), and FedFor (Tian et al., 2022). Specifically, FedAvgM and FedAdam use different server-side momentum while FedProx and FedFor have different client-side regularization. For domain shift experiments, we include the SOTA personalized FL alogorithm FedBN (Li et al., 2021). For all experiments, we assume a large amount of available clients and sample a fraction of them to participate in each round of communication. This corresponds to the most practical FL setting: cross-device FL with partial participation (Kairouz et al., 2021). For different experiments, we use different neural network architectures. Please refer to the Appendix A.3 for more details.

---

[2]Some FL algorithms require exact convergence on local model, e.g., FedPD (Zhang et al., 2020).

## 4.1 IMBALANCE EXPERIMENTS

For imbalance experiments, we use the Imbalanced CIFAR10 (Cao et al., 2019) dataset created with an artificial exponential imbalance among classes. To create this exponential imbalance, we specify a variable *imbalance ratio*. For example, an imbalance ratio of 0.01 means that the ratio between the number of samples in the smallest class and the largest class is 0.01.

Table 1: **Compatibility Experiments on Imbalanced CIFAR.** Results are averaged over 3 runs. $E$ is the number of local iterations and $t$ refers to the number of global iterations. A complete table with standard deviation is available at A.5.

| Backbone | Strategy | $E = 20$ | | | | $E = 40$ | | | | $E = 160$ | | | |
|---|---|---|---|---|---|---|---|---|---|---|---|---|---|
| | | acc40↓ | $t = 200$ ↑ | $t = 400$ ↑ | Test↑ | acc40↓ | $t = 100$ ↑ | $t = 200$ ↑ | Test↑ | acc40↓ | $t = 25$ ↑ | $t = 50$ ↑ | Test ↑ |
| FedAvg | Propto. | 43.33 | 55.89 | 64.78 | 64.60 | 22.33 | 55.77 | 64.41 | 64.25 | 7.00 | 54.85 | 63.48 | 63.55 |
| | Exp-$\alpha$ | **38.00** | **58.24** | **67.19** | **66.86** | **19.67** | **57.20** | **66.12** | **65.59** | **5.33** | **57.77** | **66.12** | **66.09** |
| FedProx | Propto. | 47.33 | 56.53 | 64.86 | 64.57 | 25.00 | 55.92 | 65.44 | 64.98 | 7.33 | 55.52 | 64.06 | 63.91 |
| | Exp-$\alpha$ | **36.00** | **57.99** | **65.74** | **65.54** | **22.00** | **57.41** | **66.22** | **65.83** | **6.33** | **56.55** | **66.05** | **65.63** |
| FedFor | Propto. | 37.67 | 60.80 | 69.78 | 69.62 | 16.33 | 64.71 | 74.04 | 73.25 | 6.33 | 57.74 | 67.02 | 66.71 |
| | Exp-$\alpha$ | **34.67** | **62.15** | **70.48** | **70.04** | 16.67 | 64.47 | **74.44** | **73.90** | **5.33** | **59.49** | **69.34** | **69.06** |
| FedAvgM | Propto. | 26.67 | 63.43 | 71.36 | 70.72 | 16.33 | 62.38 | 70.55 | 70.28 | 8.00 | 59.25 | 69.17 | 68.40 |
| | Exp-$\alpha$ | **25.00** | **65.04** | **72.65** | **72.21** | **13.00** | **64.15** | **72.69** | **72.55** | **6.00** | **61.08** | **69.61** | **69.52** |
| FedAdam | Propto. | 22.33 | 72.67 | 79.38 | 78.87 | **14.67** | 67.59 | 76.41 | 76.15 | **10.00** | **55.05** | 64.98 | 64.32 |
| | Exp-$\alpha$ | **18.67** | **74.16** | **80.56** | **79.69** | 14.33 | **69.26** | **76.86** | **76.49** | 11.33 | 54.75 | **65.37** | **65.20** |

**Compatibility to other FL algorithms.** To mimic the real world situation of varying imbalance, we sample a batch of 10 clients for each round of communication and each client is created using an imbalance ratio sampled randomly from the set of ratios $\{1.0, 0.5, 0.1, 0.05, 0.01, 0.005, 0.001\}$ covering a gradual increase of severity of imbalance from balanced, moderately imbalanced and extremely imbalanced datasets. At the beginning of each round, we re-sample the clients and their imbalance ratios. We benchmark the performance of Exp-$\alpha$ and proportional aggregation in six FL algorithms under four communication-computation configurations with a trade-off between the number of global iterations (denoted as $t$) and local update iterations (denoted as $E$) in Tab. 1. More local update (larger $E$) iterations lead to more severe weight divergence (Li et al., 2019) but with potential global communication savings (smaller $t$). Here, we report the number of steps to reach 40% accuracy (denoted as acc40), half-time, final validation accuracy, and the final test accuracy. Moreover, we fix the temperature hyperparameter $\alpha = 0.2$, chosen by a grid search on a validation set using FedAvg and FedAvgM with $\{t = 100, E = 80\}$ and an imbalance ratio of 0.001. Please see Appendix A.5 for the details on effects of $\alpha$. We observe that Exp-$\alpha$ brings improvement to all FL algorithms considered[3]. Specifically, most algorithms using Exp-$\alpha$ reach 40% accuracy with fewer rounds of global steps than using proportional aggregation, e.g., 12.3% reduction in number of steps using FedAvg with $E = 20$. This shows that Exp-$\alpha$ improves convergence rate over proportional aggregation. Furthermore, all algorithm using Exp-$\alpha$ reach higher converged accuracy, thus lower error, than using proportional aggregation. This shows that in real world settings, where function are non-convex, faster convergence speed can potentially lead to lower error, in contrast to the speed and error trade-off in theoretical convergence analysis in convex settings.

**Adaptability to Severity of Imbalance.** As a dynamic algorithm, Exp-$\alpha$ should weight each sample differently based on the severity of imbalance. To provide more insights into the adaptability to the degree of imbalance, we now use more controlled sampling strategies and imbalance ratios. Instead of sampling clients with random imbalance ratio as in the previous experiment, we designate a few combinations of imbalance configurations. For example, we use the notation $\{1.0 \times 4, 0.01 \times 3, 0.1 \times 3\}$ to denote a composition of four balanced clients, three imbalanced clients with an imbalance ratio of 0.01 and three imbalanced clients with an imbalance ratio of 0.1. Specifically, we visualize the aggregation weights of Exp-$\alpha$ and compare its performance against that of the proportional aggregation in four imbalance configurations. Again, we sample ten clients at the beginning of each round however with different imbalance configurations. The hyperparameter $\alpha$ is set to 0.2 and the FL setting is $\{t = 400, E = 20\}$. As we can see from Fig. 2a, Exp-$\alpha$ adapts to different imbalance configurations. For example, as a client becomes less imbalanced, it receives a higher

---

[3]Exp-$\alpha$ can be applied to other FL algorithms such as SCAFFOLD (Karimireddy et al., 2020), Fed-Dyn (Acar et al., 2021). However, these algorithms are not compatible with the current benchmark because they are *stateful* algorithms and perform poorly in the cross-device setting (Xu et al., 2021).

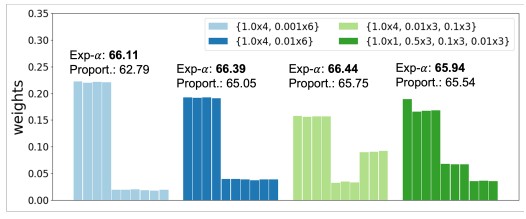 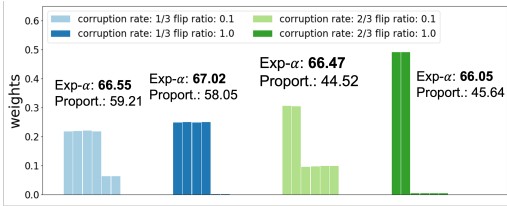

(a) Imbalance Adaptability Experiments          (b) Flip Adaptability Experiments

Figure 2: We visualize the average aggregate weights and report test accuracy of Exp-$\alpha$ for different imbalance and flip configurations. Exp-$\alpha$ always assigns smaller weight to more shifted clients. Fedavg is used as the backbone FL algorithm and results are averaged over three runs.

weight. The balanced clients always have the highest aggregation weights. Also, as expected, Exp-$\alpha$ shows more performance improvement when imbalance is more severe. In the case, when all clients are balanced, Exp-$\alpha$ assign roughly equal weights to all clients with marginal variation due to stochasticity in sampling. Experiments with IID clients are provided in Appendix A.5. As expected, when clients have balanced datasets, Exp-$\alpha$ performs just as well as proportional aggregation.

## 4.2 LABEL FLIPPING EXPERIMENTS

For label flipping experiments, we use CIFAR10 as the dataset. To control the extend of label flipping, we define *corruption rate* and *flip ratio*. For example, a corruption rate of $1/3$ means that with $1/3$ chance, a client has flipped labels and a flip ratio of 1.0 means that all classes are incorrectly labeled and 0.5 means that half of the classes are incorrectly labeled. The number of classes whose labels are flipped is determined by the flip ratio multiplied by the number of classes.

Table 2: **Compatibility Experiments on CIFAR with flipped clients.** Results are averaged over 3 runs. $E$ is the number of local iterations and $t$ refers to the number of global iterations. A complete table with standard deviation is available at A.6.

| Backbone | Strategy | $E = 20$ | | | | $E = 40$ | | | | $E = 160$ | | | |
|---|---|---|---|---|---|---|---|---|---|---|---|---|---|
| | | acc40↓ | $t = 200$ ↑ | $t = 400$ ↑ | Test↑ | acc40↓ | $t = 100$ ↑ | $t = 200$ ↑ | Test↑ | acc40↓ | $t = 25$ ↑ | $t = 50$ ↑ | Test ↑ |
| FedAvg | Propto. | 69.33 | 52.00 | 59.55 | 59.47 | 31.67 | 52.80 | 60.59 | 60.21 | 10.00 | 49.65 | 60.73 | 60.45 |
| | Exp-$\alpha$ | **36.33** | **58.39** | **66.97** | **67.01** | **18.33** | **57.19** | **65.78** | **66.21** | **5.33** | **58.34** | **66.45** | **66.24** |
| FedProx | Propto. | 59.67 | 52.25 | 60.08 | 60.27 | 33.67 | 53.03 | 61.00 | 60.83 | 12.33 | 51.11 | 59.71 | 59.71 |
| | Exp-$\alpha$ | **34.33** | **58.95** | **67.14** | **66.51** | **19.67** | **58.31** | **66.74** | **66.75** | **5.00** | **58.65** | **66.71** | **66.60** |
| FedFor | Propto. | 53.00 | 54.54 | 61.71 | 61.98 | 25.67 | 56.77 | 66.79 | 66.49 | 9.67 | 53.32 | 62.78 | 61.94 |
| | Exp-$\alpha$ | **28.00** | **62.50** | **70.89** | **70.39** | **15.33** | **64.50** | **73.02** | **72.57** | **5.33** | **60.56** | **70.18** | **69.70** |
| FedAvgM | Propto. | 39.33 | 56.11 | 64.29 | 64.13 | 28.00 | 56.95 | 65.76 | 65.82 | 10.67 | 54.45 | 61.40 | 61.05 |
| | Exp-$\alpha$ | **22.67** | **65.95** | **73.82** | **73.45** | **13.00** | **64.93** | **73.28** | **72.40** | **6.00** | **62.66** | **70.97** | **70.62** |
| FedAdam | Propto. | 30.00 | 61.61 | 71.72 | 71.51 | 24.33 | 57.46 | 64.37 | 64.00 | 18.67 | 43.53 | 57.27 | 57.07 |
| | Exp-$\alpha$ | **20.00** | **74.15** | **80.17** | **79.51** | **12.33** | **70.14** | **77.46** | **76.79** | **11.33** | **54.22** | **64.40** | **64.09** |

**Compatibility to other FL algorithms.** Similar to the imbalance experiments, we benchmark Exp-$\alpha$ and proportional aggregation using six different FL backbone algorithms. In this experiment, we keep $\alpha = 0.2$, chosen by a grid search on a validation set using FedAvg and FedAvgM with a corruption rate of $1/3$, a flip ratio of 1.0 and $\{t = 100, E = 80\}$. Please see Appendix A.6 for the effects of $\alpha$. At the beginning of each round, we sample six clients with different data composition and each client has a probability of $1/3$ being corrupted with a flip ratio of 1.0, meaning that all its labels are incorrect. Therefore, the algorithm is challenged with a different flipping pattern each time. We report the number of global steps to reach $40\%$ accuracy, half-time, final time validation accuracy and final test accuracy across four federated learning configurations in Tab. 2. In all experiments, Exp-$\alpha$ brings significant improvements over proportional aggregation. This demonstrates that 1) label flipping is detrimental to federated learning and 2) Exp-$\alpha$ can greatly alleviate its negative affect.

**Adaptability to Severity of Label Flipping.** In this experiment, we study how Exp-$\alpha$ responds to partially flipped clients. Specifically, we keep $\alpha = 0.2$ and vary the flip ratio $\in \{1.0, 0.1\}$, meaning a corrupted client can either have all labels wrong or just labels for one class incorrect. Furthermore,

instead of sampling, we use a *deterministic* corruption rate $\in \{2/3, 1/3\}$. This means that at the beginning of each round of communication, we sample six clients and four or two clients out of the six clients will be corrupted, corresponding to $2/3$ or $1/3$ corruption rate respectively. The hyperparameter $\alpha$ is set to 0.2 and the FL setting is $\{t = 400, E = 20\}$. In Fig. 2b, we present four configurations covering the aforementioned variables of interest. We notice that 1) Exp-$\alpha$ outperforms proportional aggregation in all configurations; 2) Exp-$\alpha$ differentiates between clients with different level of label flipping, e.g., it assigns higher weights to flipped clients where only a single class is incorrect than to those where all classes are incorrect. Furthermore, Exp-$\alpha$ works when the number of flipped clients is majority and minority.

### 4.3 DOMAIN SHIFT EXPERIMENTS

Table 3: **Domain Shift Experiments on Digits, Office-Clatech10 and DomainNet.** Results are averaged over 3 runs. Each benchmark has several domains. We use shorthand notation in this table.

| Backbone | Strategy | Digits (Li et al., 2021) | | | | | | Office-Caltech10 (Gong et al., 2012) | | | | | DomainNet (Peng et al., 2019) | | | | | | |
|---|---|---|---|---|---|---|---|---|---|---|---|---|---|---|---|---|---|---|---|
| | | MNIST | SVHN | USPS | SD | M-M | Avg. | A | C | D | W | Avg. | C | P | I | R | S | Q | Avg. |
| FedAvg | Propoto | **95.64** | **61.33** | 95.38 | **81.82** | 75.26 | 81.89 | 57.47 | 50.81 | **65.63** | 77.40 | 62.83 | 72.37 | 64.13 | 38.81 | 69.87 | 69.31 | 59.23 | 62.29 |
| | Exp-$\alpha$ | 95.62 | 61.13 | **95.39** | 81.77 | **75.71** | **81.92** | **59.03** | **51.70** | 64.58 | **86.44** | **65.44** | **73.44** | **65.05** | **39.93** | **70.01** | **70.76** | **60.50** | **63.28** |
| FedBN | Propoto | 90.77 | **74.66** | 96.97 | 78.88 | **84.83** | 85.22 | 76.30 | 55.78 | 92.19 | **94.92** | 79.80 | 76.56 | 69.85 | **44.55** | **81.38** | 72.75 | 82.67 | 71.29 |
| | Exp-$\alpha$ | **96.73** | 70.25 | **97.04** | **83.28** | 78.91 | **85.24** | **78.82** | **56.45** | **93.75** | 93.79 | **80.70** | **78.84** | **71.30** | 43.48 | 80.97 | **73.77** | **84.47** | **72.14** |

Unlike in the previous two challenges, for domain shift, a federated learning algorithm needs to consider all clients *despite* their data heterogeneity. We benchmark Exp-$\alpha$ and proportional aggregation on three domain shift benchmarks: Digits, Office, and DomainNet. Each benchmark consists of several domains with a shared label space. Specifically, Digits consists of five digit-like datasets; Office-Caltech10 has four domains and DomainNet has six domains. Please see Appendix A.4 for detailed description. We distribute the data from each domain to a client separately, such that each client has a distinct data distribution with domain shift. FedAvg (McMahan et al., 2017) and FedBN (Li et al., 2021) are used as the backbone FL algorithms. We report test accuracy of each domain and cross-domain average for each benchmark under the FL setting $\{t = 400, E = 16\}$ in Tab. 3. Different $\alpha$ has been used for each dataset, chosen by grid search using the validation set. Please see Appendix A.7 for the discussion on effects of $\alpha$. We observe that Exp-$\alpha$ leads to similar performance as proportional aggregation in most cases. This shows that Exp-$\alpha$ incorporates all local data despite the existence of domain shifts among clients. Furthermore, Exp-$\alpha$ even brings noticeable improvement in some cases . For example, Exp-$\alpha$ improves FedAvg with proportional aggregation on Office-Clatech10 by a relatively $4\%$ on average across four domains. Upon close examination, the improvement mainly comes from the Webcam (W) domain (relatively $10\%$ improvement). The Webcam domain is the best performing domain already when using proportional aggregation, indicating that it benefits the most from federated learning across the four domains. Exp-$\alpha$ emphasizes it further by assigning the Webcam domain the largest aggregation weight. For this particular experiment, the average aggregation weights over the entire training trajectory for Amazon (A), Caltech (C), DSLR (D) and Webcam (W) are $\{0.19, 0.26, 0.22, 0.34\}$. While Amazon (A) received the smallest average aggregation weight, this did not deteriorate the performance of Exp-$\alpha$ on this domain but rather improved it by a relative $3\%$.

## 5 CONCLUSION

While proportional aggregation enjoys several theoretical advantages, e.g., equal importance and asymptotic convergence, a fixed client weighting is less sensible under non-IID settings. In this paper, we start out by removing the assumption of proportional aggregation and derive a convergence bound using a generic and time-varying aggregation strategy. This analysis reveals a surprising trade-off between convergence speed and error under convexity assumptions. The analysis motivates a family of aggregation strategies, which prioritize convergence speed and weight samples dynamically. Consequently, we propose a new aggregation strategy, Exp-$\alpha$, from this family. Our extensive experiments on three types of data heterogeneity demonstrates its superior performance and robustness, and compatibility to existing algorithm albeit the existence of non-zero error in theory. More importantly, the theoretical analysis opens a new direction to study aggregation strategy to focus on convergence speed and robustness in future works.

## 6 REPRODUCIBILITY STATEMENT

We include a code repository to reproduce the results on imbalanced CIFAR10 reported in Tab. 1. Specifically, the codebase includes an implementation of FedAvg (McMahan et al., 2017) with the original proportional aggregation and the proposed Exp-$\alpha$ aggregation strategy. Readers can reproduce results reported in the first row of Tab. 1. The code repository has a readme file with necessary instructions to install environment and run experiments. The code is written to run on a single GPU.

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

# A APPENDIX

## A.1 EXTENDED RELATED WORKS

**Client Update.** FedProx (Zhao et al., 2018) adds a first order proximal term in the loss function; FedCurv (Shoham et al., 2019) and FedFor (Tian et al., 2022) use a second order and first order gradient regularization respectively; SCAFFOLD (Karimireddy et al., 2020) uses control-variate to align objective functions; FedPD (Zhang et al., 2020)/FedDyn (Acar et al., 2021) propose a dynamic regularization based on gradient consensus among clients.

**Server Update.** In the aggregation step, updated gradients are averaged in a *weighted* manner to generate pseudo-gradients for the global model (Reddi et al., 2020). In the optimization step, methods differ in how the pseudo-gradients are applied. FedAvg (McMahan et al., 2017) directly uses SGD-like updates; FedAvgM (Hsu et al., 2019) adds Nesterov momentum and FedAdam/FedYogi (Reddi et al., 2020) generalizes more adaptive optimization techniques. Our method specifically tackles the less discussed aggregation step.

## A.2 ALGORITHM DESCRIPTION OF EXP-$\alpha$

---

**Algorithm 1:** Exp-$\alpha$.

**Data:** $K$ clients with local data $\Xi_i$ for $i \in \{1, ..., K\}$ are selected, temperature parameter $\alpha$.
**Result:** $\mathbf{W}^T$
Initialize $\mathbf{W}^0$
**for** $T \in \{nE|n = 0, ..2\}$ **do**
$\quad$ **for** $i \in \{1, ..., K\}$ *in parallel* **do**
$\quad\quad F_i(\mathbf{W}^T) \leftarrow$ CalculateRisk $(\Xi_i, \mathbf{W}^T)$
$\quad\quad \mathbf{W}_i^{T+E} \leftarrow$ ClientUpdate$(\mathbf{W}^T)$ $\qquad\qquad\qquad$ ▷ e.g., FedProx Zhao et al. (2018)
$\quad\quad F_i(\mathbf{W}_i^{T+E}) \leftarrow$ CalculateRisk $(\Xi_i, \mathbf{W}_i^{T+E})$
$\quad\quad \rho_i^{T+E} = \exp\left(\frac{F_i(\mathbf{W}_i^{T+E}) - F_i(\mathbf{W}^T)}{\alpha}\right).$ $\qquad\qquad$ ▷ Eq. 3
$\quad$ **end**
$\quad \nabla F(\mathbf{W}^T) = \frac{1}{\sum_{j=1}^{K} \rho_j^{T+E}} \sum_{i=1}^{K} \rho_i^{T+E} \left(\mathbf{W}^T - \mathbf{W}_i^{T+E}\right)$
$\quad \mathbf{W}^{T+E} \leftarrow$ ServerUpdate$(\mathbf{W}^T, \nabla F(\mathbf{W}^T))$ $\qquad\qquad$ ▷ e.g., FedAvgM Hsu et al. (2019)
**end**

---

In this section, we describe the Exp-$\alpha$ algorithm (Alg. 1). Exp-$\alpha$ is an aggregation algorithm, so it is compatible with most existing FL algorithms. Following the convention in (Reddi et al., 2020) and to describe the algorithm as general as possible, we abstract the client optimization and global optimization procedures as *ClientUpdate* and *ServerUpdate*. The majority of FL algorithms differ in how they change these two components. Please refer to the related works section (Sec. 2) for a brief discussion on this. In our experiments (Sec. 4), we aim to demonstrate generality and compatibility of Exp-$\alpha$ in combination of innovations to these components. In Alg. 1, we implement a *CalculateRisk* function to calculate the risk values. This is a simple inference forward pass through the local dataset given specific model. The computation on the client side is fairly cheap, as it only requires two additional inference passes on the local data. During local training, an FL algorithm needs to run forward-backward pass, e.g., computing, storing and applying gradients for multiple epochs, each of which has many more local iterations while the calculation of risk only requires a simple forward pass without computing, storing and applying gradients. Therefore, the computation cost of *CalculateRisk* is only a small fraction of the original computation cost.

## A.3 IMPLEMENTATION DETAILS

For CIFAR experiment in Sec. 4.1 and 4.2, we use ResNet20 (He et al., 2016). Specifically, we use the proper ResNet implementation for CIFAR10 (He et al., 2016). For Digits experiment in Sec. 4.3, we use a custom CNN provided by Li et al. (2021). For Office and DomainNet experiment in Sec 4.3, we use ResNet18 (He et al., 2016). All models are trained with SGD, with no momentum and weight decays. We use constant learning rate, i.e., no learning rate decay: CIFAR, Digits and Office 0.01

Table 4: **Implementation Details for CIFAR10, Digits, Office and DomainNet.**

| Dataset | Architecture | Optimizer | Learning Rate | Batch Size | Steps per Epoch |
|---|---|---|---|---|---|
| CIFAR10 | ResNet20 He et al. (2016) | SGD | 0.01 | 128 | 10 |
| Office-Caltech10 | ResNet18 He et al. (2016) | SGD | 0.01 | 64 | 4 |
| DomainNet | ResNet18 He et al. (2016) | SGD | 0.05 | 64 | 4 |
| Digits | DigitModel Li et al. (2021) | SGD | 0.01 | 64 | 4 |

and DomainNet 0.05. We summarize the statistics in Tab. 4. To avoid the issue of implicit bais due to difference in number of local updates (Wang et al., 2020), we keep the optimization steps per epoch constant on all clients in one experiment. Therefore, we also report the the number of steps in each local epoch in Tab. 4 for each dataset.

## A.4 DATASET STATISTICS

Table 5: **Train, validation and test splits for the Office-Caltech10 dataset.**

| | Amazon | Caltech | DSLR | Webcam |
|---|---|---|---|---|
| Train | 536 | 628 | 87 | 165 |
| Val | 230 | 270 | 38 | 71 |
| Test | 192 | 225 | 32 | 59 |

Table 6: **Train, validation and test splits for the DomainNet dataset.**

| | Clipart | Painting | Infograph | Real | Sketch | Quickdraw |
|---|---|---|---|---|---|---|
| Train | 1472 | 1730 | 1838 | 3404 | 1549 | 2800 |
| Val | 631 | 742 | 788 | 1460 | 664 | 1200 |
| Test | 526 | 619 | 657 | 1217 | 554 | 1000 |

The Digits benchmark consists of SVHN (Netzer et al., 2011), USPS Hull (Hull, 1994), SynthDigits (Ganin & Lempitsky, 2015) and MNIST-M (Ganin & Lempitsky, 2015), MNIST (LeCun et al., 1998); the DomainNet benchmark (Peng et al., 2019) has six domains. The Office-Caltech10 dataset (Gong et al., 2012) selects three doamins from Office-31 (Saenko et al., 2010), Amazon, DSLR and Webcam, and one domain from Caltech256 (Griffin et al., 2007).

We split datasets into training, validation and test sets. In our experiments we report validation accuracy and test accuracy if applicable. We summarize the number of samples in each split for Office, DomainNet and Digits in domain shift experiments in Tab. 5,Tab. 6 and Tab. 7 respectively. For CIFAR 10 experiments, we have the following splits $\{train : 35,000, validation : 15,000, test : 10,000\}$.

## A.5 ADDITIONAL RESULTS FOR IMBALANCE EXPERIMENTS IN SEC 4.1

**Effects of Alpha.** In this experiment, we fix the imbalance ratio to 0.001, the number of local epochs as 2 and the number of global iteration 200, and vary the hyperaparameter $\alpha$ in Exp-$\alpha$. Specifically, we sample a different set of ten clients, among which four are balanced and six are imbalanced each time. So the total number available clients is the number of rounds of communication multiplied by ten. We compare convergence performance of under different $\alpha \in \{0.2, 1.0, 5.0, 25.0, 125.0\}$ using two backbone FL algorithms: FedAvg (McMahan et al., 2017) and FedavgM (Hsu et al., 2019) in Fig. 3. Compared to proportional aggregation, Exp-$\alpha$ with a proper selection of $\alpha$ can consistently improve both convergence speed and converged performance. Specifically in this experiment, we noticed that smaller $\alpha$ leads to better performance because a smaller $\alpha$ makes the weights more concentrated on the balanced clients. We use $\alpha = 0.2$ in the main paper in Sec. 4.1.

**Effects of Local Steps and Degree of Heterogeneity.** With increasing increasing number of local steps and increasing heterogeneity among clients, a smaller $\alpha$ can do better. To demonstrate this we present the following experiments. Specifically, At each round of global communication, we

Table 7: **Train, validation and test splits for the Digits dataset.**

|       | MNIST | SVHN  | USPS | SynthDitgits | MNIST-M |
|-------|-------|-------|------|--------------|---------|
| Train | 892   | 892   | 892  | 892          | 892     |
| Val   | 595   | 595   | 595  | 595          | 595     |
| Test  | 14000 | 19858 | 1860 | 97791        | 14000   |

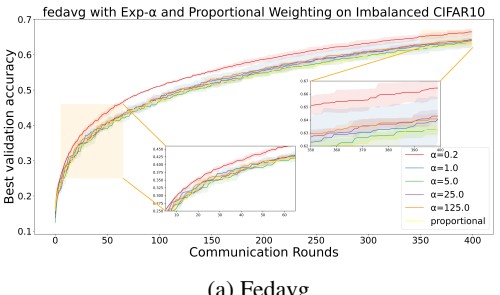 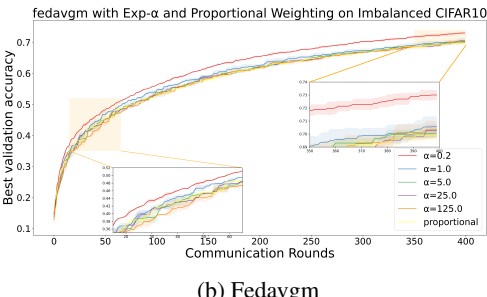

(a) Fedavg          (b) Fedavgm

Figure 3: Convergence of Fedavg and Fedavgm using Exp-$\alpha$ and proportional aggregation on Imbalance CIFAR10. Results are averaged over three runs. Imbalance ratio $= 0.001$.

sample a different set of 10 clients, six of which are imbalanced. To show the effects of increasing local steps, we run experiments across four communication settings with increasing number of local steps $E \in \{20, 40, 80, 160\}$ and a fixed imbalance ratio of 0.1. For each setting, we sweep $\alpha \in \{0.2, 1, 5, 25, 125\}$. In Tab. 8, we show the test accuracy with different $\alpha$ for each communication setting. We observe that increasing number of local steps requires smaller $\alpha$.

To show the effects of increasing heterogeneity, we fix the number of local steps to be $E = 160$ and vary the imbalance ration in $\{0.1, 0.2, 0.3, 0.4, 0.5\}$ with smaller number indicating more severe imbalance. Similarly, we sweep $\alpha \in \{0.2, 1, 5, 25, 125\}$. In Tab. 9 we present test accuracy for each imbalance ratio with different $\alpha$. We observe that more severe heterogeneity can benefit from smaller $\alpha$.

**Imbalance Compatibility Table with Standard Deviation.** Here, we show the full table of Tab. 1 with standard deviation in Tab. 10.

**Exp-$\alpha$ in IID Setting.** In the main paper, we explored Exp-$\alpha$ in non-IID settings, where clients are subject to different degrees of imbalance. In this section, we present results comparing FedAvg using Exp-$\alpha$ and proportional aggregation under IID settings, where all clients have balanced data in Tab. 11. As expected, Exp-$\alpha$ and proportional aggregation perform similarly.

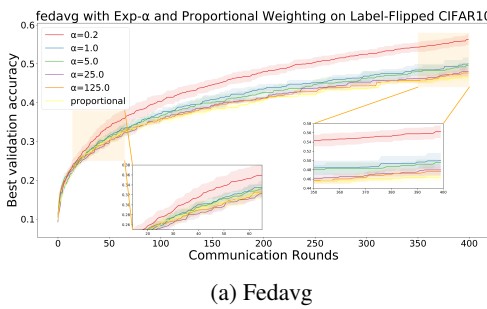 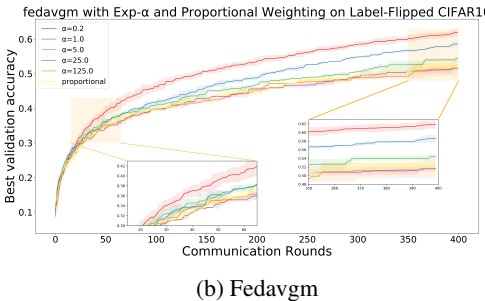

(a) Fedavg          (b) Fedavgm

Figure 4: Convergence of Fedavg and Fedavgm using Exp-$\alpha$ and proportional aggregation on Flipped CIFAR10. Results are averaged over three runs. Flip ratio $= 1.0$.

Table 8: **Effects of Local Steps with EXP-$\alpha$.** We use a fixed imbalance ratio of 0.1.

|       | 0.2   | 1.0   | 5.0   | 25.0  | 125.0 |
|-------|-------|-------|-------|-------|-------|
| E=20  | 66.68 | 66.63 | 65.35 | 66.72 | **67.13** |
| E=40  | 66.33 | 66.54 | 66.92 | 66.72 | **66.81** |
| E=80  | 65.62 | 66.61 | **66.65** | 66.37 | 66.34 |
| E=160 | **66.50** | 66.36 | 65.53 | 65.53 | 64.78 |

Table 9: **Effects of Heterogeneity with EXP-$\alpha$.** We sweep different imbalance ratio and use a fixed number of local steps $E = 160$

| Imbalance Ratio | 0.2   | 1.0   | 5.0   | 25.0  | 125.0 |
|-----------------|-------|-------|-------|-------|-------|
| 0.1             | **66.50** | 66.36 | 65.53 | 65.53 | 64.78 |
| 0.2             | **67.19** | 66.83 | 66.84 | 66.71 | 66.22 |
| 0.3             | **67.07** | 66.38 | 66.40 | 66.96 | 65.85 |
| 0.4             | 66.87 | **67.48** | 68.79 | 66.64 | 66.71 |
| 0.5             | 65.75 | 66.18 | 66.16 | 66.63 | **67.33** |

## A.6   ADDITIONAL RESULTS FOR LABEL FLIPPING EXPERIMENTS IN SEC. 4.2

**Effects of Alpha.** In this experiment, we fix the corruption rate to be $1/3$, the flip ratio to be 1.0, i.e., all labels on a flipped client are incorrect, and vary the hyperparamter $\alpha$. Specifically, we randomly sample data from a training set to create six balanced clients. However, clients are subject to label-flipping with a chance of $1/3$. We compare convergence performance of under different $\alpha \in \{0.2, 1.0, 5.0, 25.0, 125.0\}$ using two backbone FL algorithms: fedavg (McMahan et al., 2017) and fedavgm (Hsu et al., 2019) in Fig. 4. We notice that Exp-$\alpha$ with small $\alpha$ values provides significant convergence improvement compared to proportional aggregation. This is because a smaller $\alpha$ forces the aggregation algorithm to focus more on the clean clients. We use $\alpha = 0.2$ in the main paper in Sec. 4.2.

**Label Flipping Compatibility Table with Standard Deviation.** Here, we show the full table of Tab. 2 with standard deviation in Tab. 12.

## A.7   ADDITIONAL RESULTS FOR DOMAIN SHIFT EXPERIMENTS IN SEC. 4.3

**Effects of Alpha.** In this section, we sweep across a range of $\alpha \in \{0.2, 1, 5, 25, 125\}$ on the three domain shift benchmarks: Digits, Office-Caltech10 and DomainNet. We report validation accuracy in Tab. 13. While there is not obvious trend on which $\alpha$ works the best, Exp-$\alpha$ with a moderate $\alpha \geq 1$ outperforms proportional aggregation in most cases.

## A.8   LEMMAS

To facilitate the derivation of the main convergence bound, we will introduce some lemmas. Specifically, we will utilize the method of virtual sequence (Stich, 2018). Let $\mathcal{I}_E$ be the set of synchronization/communication steps, such that $\mathcal{I}_E = \{nE|n = 0, ..2\}$ where $E$ denotes the number of local update iterations. We first introduce an intermediate notation denoting the update of a single step SGD update on a client:

$$\mathbf{V}_i^{t+1} = \mathbf{W}_i^t - \eta_t \nabla f_i(\mathbf{W}_i^t; \xi_i^t). \tag{4}$$

where $\xi_i^t \sim P_i$ is a sampled data point on the client $i$ at time $t$.

Depending on whether the current iteration is a synchronization step, the local update on each client can be written as the following, for $t \geq 0$:

$$\mathbf{W}_i^{t+1} = \begin{cases} \mathbf{V}_i^{t+1} & \text{if } t+1 \notin \mathcal{I}_E, \tag{5} \\ \sum_{i=1}^{N} \rho_i^{t+1} \mathbf{V}_i^{t+1} & \text{if } t+1 \in \mathcal{I}_E. \tag{6} \end{cases}$$

where $\rho_i^{t+1} \geq 0$ and $\sum_{i=1}^{N} \rho_i^{t+1} = 1$ is the aggregation weight for client $i$, and $\mathbf{W}_i^0 = \mathbf{W}^0$.

Table 10: **Compatibility Experiments on Imbalanced CIFAR.** Results are averaged over 3 runs. $E$ is the number of local iterations and $t$ refers to the number of global iterations.

| Backbone | Strategy | E=20 acc40↓ | $t=200\uparrow$ | $t=400\uparrow$ | Test↑ | E=40 acc40↓ | $t=100\uparrow$ | $t=200\uparrow$ | Test↑ | E=80 acc40↓ | $t=50\uparrow$ | $t=100\uparrow$ | Test↑ | E=160 acc40↓ | $t=25\uparrow$ | $t=50\uparrow$ | Test↑ |
|---|---|---|---|---|---|---|---|---|---|---|---|---|---|---|---|---|---|
| FedAvg | Propto. | 43.33 ±3.21 | 55.89 ±1.44 | 64.78 ±0.60 | 64.60 ±0.34 | 22.33 ±0.58 | 55.77 ±1.11 | 64.41 ±1.62 | 64.25 ±1.44 | 11.33 ±1.53 | 56.44 ±1.49 | 64.81 ±0.58 | 64.73 ±0.62 | 7.00 ±2.65 | 54.85 ±1.35 | 63.48 ±0.19 | 63.55 ±0.67 |
| | Exp-α | 38.00 ±3.00 | 58.24 ±0.36 | 67.19 ±0.14 | 66.86 ±0.34 | 19.67 ±1.53 | 57.20 ±1.19 | 66.12 ±0.93 | 65.59 ±0.25 | 11.33 ±2.52 | 57.74 ±0.85 | 66.20 ±0.43 | 65.81 ±0.59 | 5.33 ±0.58 | 57.77 ±1.04 | 66.12 ±0.83 | 66.09 ±0.88 |
| FedProx | Propto. | 47.33 ±2.31 | 56.53 ±0.15 | 64.86 ±0.66 | 64.57 ±0.16 | 25.00 ±5.57 | 55.92 ±1.68 | 65.44 ±1.09 | 64.98 ±0.68 | 12.67 ±1.53 | 54.97 ±0.60 | 64.35 ±0.81 | 64.05 ±0.56 | 7.33 ±0.58 | 55.52 ±0.32 | 64.06 ±0.64 | 63.91 ±0.49 |
| | Exp-α | 36.00 ±1.00 | 57.99 ±0.68 | 65.74 ±0.86 | 65.54 ±0.78 | 22.00 ±1.00 | 57.41 ±0.34 | 66.22 ±0.16 | 65.83 ±0.54 | 10.00 ±1.00 | 57.90 ±0.43 | 66.53 ±0.62 | 66.08 ±0.37 | 6.33 ±1.53 | 56.55 ±0.88 | 66.05 ±0.42 | 65.63 ±0.50 |
| FedFor | Propto. | 37.67 ±3.06 | 60.80 ±0.89 | 69.78 ±0.67 | 69.62 ±0.46 | 16.33 ±1.53 | 64.71 ±1.11 | 74.04 ±1.01 | 73.25 ±1.10 | 10.33 ±0.58 | 62.91 ±0.48 | 73.52 ±0.82 | 73.34 ±0.39 | 6.33 ±0.58 | 57.74 ±1.29 | 67.02 ±0.46 | 66.71 ±0.61 |
| | Exp-α | 34.67 ±2.52 | 62.15 ±1.17 | 70.48 ±0.72 | 70.04 ±0.70 | 16.67 ±0.58 | 64.47 ±0.22 | 74.44 ±0.57 | 73.90 ±0.69 | 9.00 ±0.00 | 63.98 ±0.46 | 73.68 ±0.87 | 73.29 ±0.64 | 5.33 ±0.58 | 59.49 ±0.55 | 69.34 ±0.15 | 69.06 ±0.30 |
| FedAvgM | Propto. | 26.67 ±3.06 | 63.43 ±0.33 | 71.36 ±0.39 | 70.72 ±0.55 | 16.33 ±2.52 | 62.38 ±0.91 | 70.55 ±0.65 | 70.28 ±0.66 | 10.33 ±1.53 | 61.37 ±0.95 | 69.41 ±0.45 | 69.32 ±0.50 | 8.00 ±1.00 | 59.25 ±0.88 | 69.17 ±.94 | 68.40 ±1.20 |
| | Exp-α | 25.00 ±2.00 | 65.04 ±0.20 | 72.65 ±0.16 | 72.21 ±0.48 | 13.00 ±1.00 | 64.15 ±1.06 | 72.69 ±0.82 | 72.55 ±1.07 | 8.33 ±1.53 | 64.29 ±0.62 | 72.77 ±0.98 | 72.77 ±1.07 | 6.00 ±1.00 | 61.08 ±0.03 | 69.61 ±0.96 | 69.52 ±0.73 |
| FedAdam | Propto. | 22.33 ±5.77 | 72.67 ±1.17 | 79.38 ±0.91 | 78.87 ±0.83 | 14.67 ±0.58 | 67.59 ±1.09 | 76.41 ±0.88 | 76.15 ±0.62 | 12.00 ±1.00 | 63.03 ±1.24 | 72.01 ±0.05 | 71.50 ±0.34 | 10.00 ±2.65 | 55.05 ±1.17 | 64.98 ±0.79 | 64.32 ±1.26 |
| | Exp-α | 18.67 ±2.08 | 74.16 ±0.84 | 80.56 ±0.31 | 79.69 ±0.19 | 14.33 ±2.31 | 69.26 ±0.86 | 76.86 ±0.29 | 76.49 ±0.55 | 13.67 ±1.15 | 62.04 ±0.70 | 71.91 ±0.70 | 71.63 ±0.80 | 11.33 ±1.53 | 54.75 ±1.28 | 65.37 ±0.28 | 65.20 ±0.93 |

Table 11: **Exp-α and Proportional Aggregation on IID CIFAR.** Results are averaged over 3 runs. $E$ is the number of local iterations and $t$ refers to the number of global iterations.

| Strategy | Global Iter. | E=20 acc40↓ | $t=200\uparrow$ | $t=400\uparrow$ | Test↑ | E=40 acc40↓ | $t=100\uparrow$ | $t=200\uparrow$ | Test↑ | E=80 acc40↓ | $t=50\uparrow$ | $t=100\uparrow$ | Test↑ | E=160 acc40↓ | $t=25\uparrow$ | $t=50\uparrow$ | Test↑ |
|---|---|---|---|---|---|---|---|---|---|---|---|---|---|---|---|---|---|
| FedAvg | Propto. | 32.67 ±4.93 | 59.72 ±0.97 | 67.70 ±0.63 | 67.56 ±0.78 | 16.67 ±0.58 | 59.26 ±0.29 | 67.64 ±0.36 | 67.69 ±0.24 | 9.00 ±1.00 | 59.01 ±1.31 | 67.50 ±0.57 | 66.89 ±0.54 | 5.00 ±0.00 | 58.76 ±1.28 | 67.46 ±1.06 | 66.96 ±0.50 |
| | Exp-α | 34.00 ±3.00 | 59.30 ±1.29 | 67.59 ±1.26 | 67.17 ±1.00 | 17.00 ±2.65 | 58.48 ±1.30 | 67.06 ±0.85 | 67.30 ±0.95 | 9.00 ±1.00 | 58.94 ±0.75 | 67.30 ±0.67 | 66.88 ±0.56 | 4.67 ±0.58 | 59.52 ±0.41 | 67.85 ±0.56 | 67.66 ±0.38 |

Then we define a virtual sequence

$$\bar{\mathbf{W}}^{t+1} = \sum_{i=1}^{N} \rho_i^{t+1} \mathbf{W}_i^{t+1}$$

. In reality, we only have access to $\bar{\mathbf{W}}^{t+1}$ when $t+1 \in \mathcal{I}_E$. We provide a graphical illustration of virtual sequence with $E=3$ and two clients in Fig. 1. The virtual sequence $\bar{\mathbf{W}}^{t+1}$ can be viewed as a virtual single step SGD update from $\bar{\mathbf{W}}^t$, i.e, $\bar{\mathbf{W}}^{t+1} = \bar{\mathbf{W}}^t - \eta_t g_t$ where $g_t = \sum_{i=1}^{N} \rho_i^{t+1} \nabla f_i(\mathbf{W}_i^t; \xi_i^t)$. Furthermore, the expected gradient is denoted as $\bar{g}_t = \mathbb{E}[g_t] = \sum_{i=1}^{N} \rho_i^{t+1} \nabla F_i(\mathbf{W}_i^t)$. Note that Lemma 2 and Lemma 3 are adaptations of lemmas from Li et al. (2019) with the addition of time-varying aggregation weights. Deferred proof of lemmas is in A.10.

**Lemma 1** *Results of one-step SGD on the virtual sequence. With Assumption 1 and 2, if $\eta_t \leq \frac{1}{4L}$, we can show that*

$$\|\bar{\mathbf{W}}^{t+1} - \mathbf{W}^*\|^2 \leq (1 - \eta_t \mu)\|\bar{\mathbf{W}}^t - \mathbf{W}^*\|^2 + 2\sum_{i=1}^{N} \rho_i^{t+1} \left(\|\bar{\mathbf{W}}^t - \mathbf{W}_i^t\|^2\right)$$
$$+ 6L\eta_t^2 \Omega + 2\eta_t(\Gamma - \Omega_t) + \eta_t^2 \|\bar{\mathbf{g}}_t - \mathbf{g}_t\|^2.$$

*where* $\Gamma = \max_t \sum_{i=1}^{N} \rho_i^{t+1} \left(F_i(\mathbf{W}^*) - F_i(\mathbf{W}_i^*)\right)$ *and* $\Omega = \min_t \sum_{i=1}^{N} \rho_i^{t+1}(F_i(\bar{\mathbf{W}}^t) - F_i(\mathbf{W}_i^*))$.

**Lemma 2** *Bounded variance. With assumption 3, it follows that*

$$\mathbb{E}\|\mathbf{g}_t - \bar{\mathbf{g}}_t\|^2 \leq \sum_i^{N} (\rho_i^{t+1})^2 \sigma_i^2 \leq \max_t \sum_{i=1}^{N} (p_i^{t+1})^2 \sigma_i^2.$$

**Lemma 3** *Bounded divergence. With assumption 4, $\eta_t$ is non-decreasing and $\eta_t \leq 2\eta_{t+E}$, $\forall t \geq 0$, it follows that*

$$\mathbb{E}\left[\sum_{i=1}^{N} \rho_i^{t+1} \|\bar{\mathbf{W}}^t - \mathbf{W}_i^t\|^2\right] \leq 4\eta_t^2(E-1)G^2$$

Table 12: **Compatibility Experiments on CIFAR with flipped clients.** Results are averaged over 3 runs. $E$ is the number of local iterations and $t$ refers to the number of global iterations.

| Backbone | Strategy | $E=20$ | | | | $E=40$ | | | | $E=80$ | | | | $E=160$ | | | |
|---|---|---|---|---|---|---|---|---|---|---|---|---|---|---|---|---|---|
| | | acc40↓ | $t=200$↑ | $t=400$↑ | Test↑ | acc40↓ | $t=100$↑ | $t=200$↑ | Test↑ | acc40↓ | $t=50$↑ | $t=100$↑ | Test↑ | acc40↓ | $t=25$↑ | $t=50$↑ | Test↑ |
| FedAvg | Propto. | 69.33 ±3.06 | 52.00 ±2.11 | 59.55 ±1.52 | 59.47 ±1.91 | 31.67 ±2.08 | 52.80 ±0.19 | 60.59 ±0.59 | 60.21 ±0.84 | 20.00 ±3.46 | 51.40 ±0.29 | 60.39 ±0.48 | 60.12 ±1.05 | 10.00 ±1.73 | 49.65 ±1.70 | 60.73 ±0.93 | 60.45 ±0.70 |
| | Exp-$\alpha$ | 36.33 ±2.31 | 58.39 ±1.09 | 66.97 ±0.72 | 67.01 ±0.23 | 18.33 ±1.15 | 57.19 ±1.11 | 65.78 ±0.99 | 66.21 ±0.58 | 10.33 ±0.58 | 58.47 ±1.13 | 66.57 ±0.90 | 66.80 ±0.64 | 5.33 ±0.58 | 58.34 ±0.50 | 66.45 ±0.76 | 66.24 ±0.79 |
| FedProx | Propto. | 59.67 ±4.16 | 52.25 ±0.79 | 60.08 ±1.28 | 60.27 ±1.31 | 33.67 ±4.93 | 53.03 ±1.50 | 61.00 ±0.50 | 60.83 ±0.28 | 15.67 ±4.04 | 51.56 ±1.47 | 60.74 ±1.80 | 60.47 ±2.08 | 12.33 ±1.53 | 51.11 ±2.01 | 59.71 ±0.72 | 59.71 ±0.98 |
| | Exp-$\alpha$ | 34.33 ±0.58 | 58.95 ±0.38 | 67.14 ±0.27 | 66.51 ±0.44 | 19.67 ±0.58 | 58.31 ±0.40 | 66.74 ±0.24 | 66.75 ±0.64 | 9.33 ±1.15 | 58.80 ±1.01 | 66.76 ±0.29 | 66.46 ±0.14 | 5.00 ±1.00 | 58.65 ±0.96 | 66.71 ±1.22 | 66.60 ±1.21 |
| FedFor | Propto. | 53.00 ±12.00 | 54.54 ±0.92 | 61.71 ±1.17 | 61.98 ±1.15 | 25.67 ±3.79 | 56.77 ±0.41 | 66.79 ±0.29 | 61.49 ±0.61 | 16.33 ±1.53 | 57.92 ±1.15 | 66.88 ±0.36 | 66.41 ±0.83 | 9.67 ±0.58 | 53.32 ±1.55 | 62.78 ±1.34 | 61.94 ±1.50 ; |
| | Exp-$\alpha$ | 28.00 ±3.00 | 62.50 ±0.96 | 70.89 ±0.80 | 70.39 ±1.38 | 15.33 ±0.58 | 64.50 ±0.22 | 73.02 ±0.17 | 72.57 ±0.34 | 9.00 ±1.00 | 64.21 ±0.13 | 74.16 ±1.18 | 73.76 ±0.84 | 5.33 ±0.58 | 60.56 ±1.28 | 70.18 ±0.44 | 69.70 ±0.65 |
| FedAvgM | Propto. | 39.33 ±3.79 | 56.11 ±0.87 | 64.29 ±0.78 | 64.13 ±0.48 | 28.00 ±9.54 | 56.95 ±1.92 | 65.76 ±0.85 | 65.82 ±1.42 | 17.67 ±2.08 | 52.13 ±1.08 | 63.68 ±2.56 | 63.22 ±2.45 | 10.67 ±1.53 | 54.45 ±1.40 | 61.40 ±2.14 | 61.05 ±2.27 |
| | Exp-$\alpha$ | 22.67 ±2.08 | 65.95 ±0.19 | 73.82 ±0.36 | 73.45 ±0.78 | 13.00 ±1.00 | 64.93 ±0.37 | 73.28 ±0.57 | 72.40 ±0.38 | 8.33 ±1.53 | 64.74 ±0.61 | 73.05 ±0.50 | 72.46 ±0.50 | 6.00 ±1.00 | 62.66 ±0.62 | 70.97 ±0.77 | 70.62 ±0.98 |
| FedAdam | Propto. | 30.00 ±4.36 | 61.61 ±0.31 | 71.72 ±0.98 | 71.51 ±0.94 | 24.33 ±3.06 | 57.46 ±1.02 | 64.37 ±0.96 | 64.00 ±1.00 | 22.33 ±2.52 | 52.92 ±3.00 | 61.96 ±1.32 | 61.95 ±1.77 | 18.67 ±5.51 | 43.53 ±2.92 | 57.27 ±.74 | 57.07 ±0.77 |
| | Exp-$\alpha$ | 20.00 ±2.65 | 74.15 ±0.90 | 80.17 ±0.44 | 79.51 ±0.41 | 12.33 ±1.53 | 70.14 ±0.45 | 77.46 ±0.16 | 76.79 ±0.04 | 12.00 ±1.00 | 62.40 ±0.11 | 71.42 ±0.75 | 70.88 ±1.11 | 11.33 ±1.53 | 54.22 ±2.04 | 64.40 ±0.61 | 64.09 ±0.97 |

Table 13: **EXP-$\alpha$ with Varying $\alpha$ on Digits, Office-Clatech10 and DomainNet.** Results are averaged over 3 runs. We use two FL algorithms FedAvg (McMahan et al., 2017) and Fedbn (Li et al., 2021). The numbers reported are validation accuracy.

| Dataset | FL Algorithm | EXP-$\alpha$ | | | | | Proportional |
|---|---|---|---|---|---|---|---|
| | | $\alpha=0.2$ | $\alpha=1.0$ | $\alpha=5.0$ | $\alpha=25.0$ | $\alpha=125.0$ | |
| Digits (Li et al., 2021) | FedAvg | 78.41 | 81.45 | 81.45 | 81.79 | 81.65 | 81.64 |
| | Fedbn | 84.92 | 85.10 | 85.03 | 85.02 | 85.01 | 84.97 |
| Office-Clatech10 (Gong et al., 2012) | FedAvg | 63.09 | 66.58 | 65.58 | 66.07 | 65.24 | 66.10 |
| | Fedbn | 78.86 | 81.86 | 80.88 | 80.95 | 81.62 | 79.81 |
| DomainNet (Peng et al., 2019) | FedAvg | 60.50 | 61.86 | 62.01 | 61.32 | 61.65 | 61.18 |
| | Fedbn | 70.34 | 69.76 | 69.30 | 69.88 | 69.53 | 69.98 |

## A.9 PROOF OF THEOREM 1

Here we present the proof of the main theorem using the lemmas from the previous section. It follows closely the method in Li et al. (2019). While we do not claim novelty in the methodology of this derivation, we show that there exists an error term due to time-varying weighting, that has been previously ignored.

Let $\Delta_t = \mathbb{E}\|\bar{\mathbf{W}}^t - \mathbf{W}^*\|^2$. From Lemma 1, Lemma 2 and Lemma 3, we have that

$$\Delta_{t+1} \leq (1 - \eta_t \mu)\Delta_t + \eta_t^2 B \tag{7}$$

where $B = \max_t \sum_{i=1}^{N} (\rho_i^{t+1})^2 \sigma_i^2 + 8(E-1)G^2 + 6L\Omega + \frac{2}{\eta_t}(\Gamma - \Omega)$. Following the setting in Li,2019 (Li et al., 2019), we set $\eta_t = \frac{\beta}{t+\gamma}$ for some $\beta \geq \frac{1}{\mu}$ and $\gamma > 0$ such that $\eta_1 \leq \frac{1}{4L}$ and $\eta_t \leq 2\eta_{t+E}$. Let $v = \max\{\frac{\beta^2 B}{\beta\mu - 1}, \gamma\Delta_0\}$. We first assume that $\Delta_t \leq \frac{v}{\gamma + t}$ and prove by induction that this holds for all $t$. By induction,

$$\Delta_{t+1} \leq (1 - \eta_t \mu)\Delta_t + \eta_t^2 B = \left(1 - \frac{\beta}{t+\gamma}\mu\right)\Delta_t + \frac{\beta^2}{(t+\gamma)^2}B \tag{8}$$

$$\leq \left(1 - \frac{\beta}{t+\gamma}\mu\right)\frac{v}{\gamma+t} + \frac{\beta^2}{(t+\gamma)^2}B = \frac{t+\gamma-1-\beta\mu+1}{(t+\gamma)^2}v + \frac{\beta^2}{(t+\gamma)^2}B$$

$$= \frac{t+\gamma-1}{(t+\gamma)^2}v + \left[\frac{\beta^2}{(t+\gamma)^2}B - \frac{\beta\mu-1}{(t+\gamma)^2}v\right]$$

$$\leq \frac{v}{t+\gamma+1}$$

By the $\mathcal{L}$-smoothness assumption (Assump. 1),

$$\mathbb{E}[F(\bar{\mathbf{W}}^t)] - F(\mathbf{W}^*) \leq \frac{L}{2}\mathbb{E}\|\bar{\mathbf{W}}^T - \mathbf{W}^*\|^2 \leq \frac{L}{2}\frac{v}{\gamma+t} \tag{9}$$

Following Li,2019 (Li et al., 2019), we choose $\beta = \frac{2}{\mu}$, $\gamma = \max\{8\frac{L}{\mu}, E\} - 1$, and let $\eta_t = \frac{2}{\mu}\frac{1}{\gamma+t}$, we can show that the learning rate satisfies $\eta_t \leq 2\eta_{t+E}, \forall t \geq 0$. Then,

$$v = \max\left\{\frac{\beta^2 B}{\beta\mu - 1}, \gamma\Delta_0\right\} \leq \frac{\beta^2 B}{\beta\mu - 1} + \gamma\Delta_0 \leq \frac{4B}{\mu^2} + \gamma\Delta_0. \tag{10}$$

Finally, plugging this in to Eq. 9, we obtain a convergence bound as,

$$\mathbb{E}[F(\bar{\mathbf{W}}^t)] - F(\mathbf{W}^*) \leq \frac{L}{2}\frac{1}{\gamma+t}\left[\frac{4B}{\mu^2} + \gamma\Delta_0\right] \tag{11}$$

$$= \frac{L}{(\gamma+t)}\left\{\frac{2}{\mu^2}\left[\max_t\sum_{i=1}^{N}(\rho_i^{t+1})^2\sigma_i^2 + 8(E-1)G^2 + 6L\Omega + \frac{2}{\eta_t}(\Gamma - \Omega)\right] + \frac{\gamma}{2}\Delta_0\right\}$$

$$= \frac{L}{(\gamma+t)}\left(\frac{2\bar{B}}{\mu^2} + \frac{\gamma}{2}\Delta_0\right) + \frac{L}{\mu}(\Gamma - \Omega)$$

where $\Delta_0 = \|\mathbf{W}^0 - \mathbf{W}^*\|_2^2$, $\bar{B} = \max_t \sum_{i=1}^{N}(\rho_i^{t+1})^2\sigma_i^2 + 8(E-1)G^2 + 6L\Omega$, $\Gamma = \max_t \sum_{i=1}^{N}\rho_i^{t+1}(F_i(\mathbf{W}^*) - F_i(\mathbf{W}_i^*))$ and $\Omega = \min_t \sum_{i=1}^{N}\rho_i^{t+1}(F_i(\bar{\mathbf{W}}^t) - F_i(\mathbf{W}_i^*))$.

## A.10 PROOF OF LEMMAS

*Proof of Lemma 1.*

From the definition of $\bar{\mathbf{W}}^{t+1} = \bar{\mathbf{W}}^t - \eta_t\mathbf{g}_t$, we can decompose $\|\bar{\mathbf{W}}^{t+1} - \mathbf{W}^*\|^2$ as

$$\|\bar{\mathbf{W}}^{t+1} - \mathbf{W}^*\|^2 = \|\bar{\mathbf{W}}^t - \eta_t\mathbf{g}_t - \mathbf{W}^* - \eta_t\bar{\mathbf{g}}_t + \eta_t\bar{\mathbf{g}}_t\|^2 \tag{12}$$

$$= \underbrace{\|\bar{\mathbf{W}}^t - \mathbf{W}^* - \eta_t\bar{\mathbf{g}}_t\|^2}_{A_1} + \underbrace{2\eta_t\left\langle\bar{\mathbf{W}}^t - \mathbf{W}^* - \eta_t\bar{\mathbf{g}}_t, \bar{\mathbf{g}}_t - \mathbf{g}_t\right\rangle}_{A_2} + \eta_t^2\|\bar{\mathbf{g}}_t - \mathbf{g}_t\|^2.$$

In the above expression, $\mathbb{E}[A_2] = 0$ so we only need to bound $A_1$.

$$A_1 = \|\bar{\mathbf{W}}^t - \mathbf{W}^* - \eta_t\bar{\mathbf{g}}_t\|^2 = \|\bar{\mathbf{W}}^t - \mathbf{W}^*\|^2 - \underbrace{2\eta_t\left\langle\bar{\mathbf{W}}^t - \mathbf{W}^*, \bar{\mathbf{g}}_t\right\rangle}_{B_1} + \underbrace{\eta_t^2\|\bar{\mathbf{g}}_t\|^2}_{B_2}. \tag{13}$$

We first focus on $B_2$. From the $\mathcal{L}$-smooth assumption (Assump. 1), we have that

$$\|\nabla F_i(\mathbf{W}_i^t)\|^2 \leq 2L(F_i(\mathbf{W}_i^t) - F_i(\mathbf{W}_i^*)). \tag{14}$$

We now bound $B_2$ as the following,

$$B_2 = \eta_t^2\|\bar{\mathbf{g}}_t\|^2 = \eta_t^2\left\|\sum_{i=1}^{N}\rho_i^{t+1}\nabla F_i(\mathbf{W}_i^t)\right\|^2 \leq \eta_t^2\sum_{i=1}^{N}\rho_i^{t+1}\|\nabla F_i(\mathbf{W}_i^t)\|^2 \tag{15}$$

$$\leq 2L\eta_t^2\sum_{i=1}^{N}\rho_i^{t+1}(F_i(\mathbf{W}_i^t) - F_i(\mathbf{W}_i^*)).$$

where the first inequality comes from the convexity of norms and Jensen's inequality for convex functions.

To bound $B_1$, we first split it into two terms by the linearity of inner product as

$$B_1 = -2\eta_t\left\langle\bar{\mathbf{W}}^t - \mathbf{W}^*, \bar{\mathbf{g}}_t\right\rangle = -2\eta_t\left\langle\bar{\mathbf{W}}^t - \mathbf{W}^* + \mathbf{W}_i^t - \mathbf{W}_i^t, \sum_{i=1}^{N}\rho_i^{t+1}\nabla F_i(\mathbf{W}_i^t)\right\rangle \tag{16}$$

$$= 2\eta_t\sum_{i=1}^{N}\rho_i^{t+1}\underbrace{\left\langle\mathbf{W}_i^t - \bar{\mathbf{W}}^t, \nabla F_i(\mathbf{W}_i^t)\right\rangle}_{B_{1,1}} + 2\eta_t\sum_{i=1}^{N}\rho_i^{t+1}\underbrace{\left\langle\mathbf{W}^* - \mathbf{W}_i^t, \nabla F_i(\mathbf{W}_i^t)\right\rangle}_{B_{1,2}}.$$

To bound $B_{1,1}$, we invoke Cauchy-Schwarz and AM-GM inequality as the following,

$$B_{1,1} = \left\langle \mathbf{W}_i^t - \bar{\mathbf{W}}^t, \nabla F_i(\mathbf{W}_i^t) \right\rangle \leq \sqrt{\frac{1}{\eta_t}\|\bar{\mathbf{W}}^t - \mathbf{W}_i^t\|^2 \eta_t \|\nabla F_i(\mathbf{W}_i^t)\|^2} \qquad (17)$$

$$\leq \frac{1}{2}\left(\frac{1}{\eta_t}\|\bar{\mathbf{W}}^t - \mathbf{W}_i^t\|^2 + \eta_n \|\nabla F_i(\mathbf{W}_i^t)\|^2\right).$$

To bound $B_{1,2}$, we use the convexity assumption (Assump. 2), which gives

$$B_{1,2} = \left\langle \mathbf{W}^* - \mathbf{W}_i^t, \nabla F_i(\mathbf{W}_i^t) \right\rangle \leq F_i(\mathbf{W}^*) - F_i(\mathbf{W}_i^t) - \frac{\mu}{2}\|\mathbf{W}^* - \mathbf{W}_i^t\|^2. \qquad (18)$$

Now we plug Eq. 15, 16, 17 and 18 back into $A_1$ (Eq. 13) as the following,

$$A_1 = \|\bar{\mathbf{W}}^t - \mathbf{W}^* - \eta_t \bar{\mathbf{g}}_t\|^2 \leq \|\bar{\mathbf{W}}^t - \mathbf{W}^*\|^2 + \eta_t \sum_{i=1}^N \rho_i^{t+1}\left(\frac{1}{\eta_t}\|\bar{\mathbf{W}}^t - \mathbf{W}_i^t\|^2 + \eta_n\|\nabla F_i(\mathbf{W}_i^t)\|^2\right)$$

$$+ 2\eta_t \sum_{i=1}^N \rho_i^{t+1}\left(F_i(\mathbf{W}^*) - F_i(\mathbf{W}_i^t) - \frac{\mu}{2}\|\mathbf{W}^* - \mathbf{W}_i^t\|^2\right)$$

$$+ 2L\eta_t^2 \sum_{i=1}^N \rho_i^{t+1}(F_i(\mathbf{W}_i^t) - F_i(\mathbf{W}_i^*))$$

$$= \|\bar{\mathbf{W}}^t - \mathbf{W}^*\|^2 - \eta_t\mu \sum_{i=1}^N \rho_i^{t+1}\|\mathbf{W}^* - \mathbf{W}_i^t\|^2 + \sum_{i=1}^N \rho_i^{t+1}\left(\|\bar{\mathbf{W}}^t - \mathbf{W}_i^t\|^2\right)$$

$$+ 2L\eta_t^2 \sum_{i=1}^N \rho_i^{t+1}(F_i(\mathbf{W}_i^t) - F_i(\mathbf{W}_i^*)) + \eta_t^2 \sum_{i=1}^N \rho_i^{t+1}\|\nabla F_i(\mathbf{W}_i^t)\|^2$$

$$+ 2\eta_t \sum_{i=1}^N \rho_i^{t+1}\left(F_i(\mathbf{W}^*) - F_i(\mathbf{W}_i^t)\right)$$

$$\leq (1 - \eta_t\mu)\|\bar{\mathbf{W}}^t - \mathbf{W}^*\|^2 + \sum_{i=1}^N \rho_i^{t+1}\left(\|\bar{\mathbf{W}}^t - \mathbf{W}_i^t\|^2\right)$$

$$+ \underbrace{4L\eta_t^2 \sum_{i=1}^N \rho_i^{t+1}(F_i(\mathbf{W}_i^t) - F_i(\mathbf{W}_i^*)) + 2\eta_t \sum_{i=1}^N \rho_i^{t+1}\left(F_i(\mathbf{W}^*) - F_i(\mathbf{W}_i^t)\right)}_{C}.$$

The last inequality uses the convexity of norms, Jensen's inequality and Eq. 14. We now rearrange $C$.

$$C = 4L\eta_t^2 \sum_{i=1}^{N} \rho_i^{t+1}(F_i(\mathbf{W}_i^t) - F_i(\mathbf{W}_i^*)) + 2\eta_t \sum_{i=1}^{N} \rho_i^{t+1}\left(F_i(\mathbf{W}^*) - F_i(\mathbf{W}_i^t)\right) \tag{19}$$

$$+ 2\eta_t \sum_{i=1}^{N} \rho_i^{t+1} F_i(\mathbf{W}_i^*) - 2\eta_t \sum_{i=1}^{N} \rho_i^{t+1} F_i(\mathbf{W}_i^*)$$

$$= -2\eta_t \sum_{i=1}^{N} \rho_i^{t+1} F_i(\mathbf{W}_i^t) + 2\eta_t \sum_{i=1}^{N} \rho_i^{t+1} F_i(\mathbf{W}_i^*) + 4L\eta_t^2 \sum_{i=1}^{N} \rho_i^{t+1}(F_i(\mathbf{W}_i^t) - F_i(\mathbf{W}_i^*))$$

$$+ 2\eta_t \sum_{i=1}^{N} \rho_i^{t+1} F_i(\mathbf{W}^*) - 2\eta_t \sum_{i=1}^{N} \rho_i^{t+1} F_i(\mathbf{W}_i^*)$$

$$= -2\eta_t(1 - 2L\eta_t) \sum_{i=1}^{N} \rho_i^{t+1}(F_i(\mathbf{W}_i^t) - F_i(\mathbf{W}_i^*)) + 2\eta_t \sum_{i=1}^{N} \rho_i^{t+1}(F_i(\mathbf{W}^*) - F_i(\mathbf{W}_i^*))$$

$$= -\gamma_t \underbrace{\sum_{i=1}^{N} \rho_i^{t+1}(F_i(\mathbf{W}_i^t) - F_i(\mathbf{W}^*))}_{D} + 4L\eta_t^2 \Gamma_t$$

where we define $\gamma_t = 2\eta_t(1 - 2L\eta_t)$ and $\Gamma_t = \sum_{i=1}^{N} \rho_i^{t+1}(F_i(\mathbf{W}^*) - F_i(\mathbf{W}_i^*))$.

To bound $D$, we use first use the convexity assumption (Assump. 2).

$$D = \sum_{i=1}^{N} \rho_i^{t+1}(F_i(\mathbf{W}_i^t) - F_i(\mathbf{W}^*)) = \sum_{i=1}^{N} \rho_i^{t+1}(F_i(\mathbf{W}_i^t) - F_i(\bar{\mathbf{W}}^t)) + \sum_{i=1}^{N} p\rho_i^{t+1}(F_i(\bar{\mathbf{W}}^t) - F_i(\mathbf{W}^*))$$
$$\tag{20}$$

$$\geq \sum_{i=1}^{N} \rho_i^{t+1} \left\langle \mathbf{W}_i^t - \bar{\mathbf{W}}^t, \nabla F_i(\bar{\mathbf{W}}^t) \right\rangle + \sum_{i=1}^{N} \rho_i^{t+1}(F_i(\bar{\mathbf{W}}^t) - F_i(\mathbf{W}^*))$$

$$\geq \frac{1}{2} \sum_{i=1}^{N} \rho_i^{t+1} \left[ \eta_t \|\nabla F_i(\bar{\mathbf{W}}^t)\|^2 + \frac{1}{\eta_t} \|\mathbf{W}_i^t - \bar{\mathbf{W}}^t\|^2 \right] + \sum_{i=1}^{N} \rho_i^{t+1}(F_i(\bar{\mathbf{W}}^t) - F_i(\mathbf{W}^*))$$

$$\geq \frac{1}{2} \sum_{i=1}^{N} \rho_i^{t+1} \left[ 2L\eta_t(F_i(\bar{\mathbf{W}}^t) - F_i(\mathbf{W}_i^*)) + \frac{1}{\eta_t} \|\mathbf{W}_i^t - \bar{\mathbf{W}}^t\|^2 \right] + \sum_{i=1}^{N} \rho_i^{t+1}(F_i(\bar{\mathbf{W}}^t) - F_i(\mathbf{W}^*))$$

where the second last inequality uses the AM-GM inequality and the last equality comes from the $\mathcal{L}$-smooth assumption (Assump. 1).

Therefore,

$$C \leq \gamma_t \sum_{i=1}^{N} \rho_i^{t+1} \left[ L\eta_t(F_i(\bar{\mathbf{W}}^t) - F_i(\mathbf{W}_i^*)) + \frac{1}{2\eta_t}\|\mathbf{W}_i^t - \bar{\mathbf{W}}^t\|^2 \right] \tag{21}$$

$$- \gamma_t \sum_{i=1}^{N} \rho_i^{t+1}(F_i(\bar{\mathbf{W}}^t) - F_i(\mathbf{W}^*)) + 4L\eta_t^2\Gamma_t$$

$$= \gamma_t \sum_{i=1}^{N} \rho_i^{t+1} \left[ L\eta_t(F_i(\bar{\mathbf{W}}^t) - F_i(\mathbf{W}_i^*)) + \frac{1}{2\eta_t}\|\mathbf{W}_i^t - \bar{\mathbf{W}}^t\|^2 \right]$$

$$- \gamma_t \sum_{i=1}^{N} \rho_i^{t+1}(F_i(\bar{\mathbf{W}}^t) - F_i(\mathbf{W}^*) + F_i(\mathbf{W}_i^*) - F_i(\mathbf{W}_i^*)) + 4L\eta_t^2\Gamma_t$$

$$= \gamma_t(\eta_t L - 1) \sum_{i=1}^{N} \rho_i^{t+1}(F_i(\bar{\mathbf{W}}^t) - F_i(\mathbf{W}_i^*)) + (4L\eta_t^2 + \gamma_t)\Gamma_t + \frac{\gamma_t}{2\eta_t} \sum_{i=1}^{N} \rho_i^{t+1}\|\mathbf{W}_i^t - \bar{\mathbf{W}}^t\|^2$$

$$\leq \gamma_t(\eta_t L - 1) \underbrace{\min_t \sum_{i=1}^{N} \rho_i^{t+1}(F_i(\bar{\mathbf{W}}^t) - F_i(\mathbf{W}_i^*))}_{\Omega \geq 0} + 2\eta_t \underbrace{\max_t \Gamma_t}_{\Gamma \geq 0} + \frac{\gamma_t}{2\eta_t} \sum_{i=1}^{N} \rho_i^{t+1}\|\mathbf{W}_i^t - \bar{\mathbf{W}}^t\|^2$$

$$= (6\eta_t^2 L - 2\eta_t - 4\eta_t^3 L^2)\Omega + 2\eta_t\Gamma + \frac{\gamma_t}{2\eta_t} \sum_{i=1}^{N} \rho_i^{t+1}\|\mathbf{W}_i^t - \bar{\mathbf{W}}^t\|^2$$

$$\leq 6\eta_t^2 L\Omega + 2\eta_t(\Gamma - \Omega) + \sum_{i=1}^{N} \rho_i^{t+1}\|\mathbf{W}_i^t - \bar{\mathbf{W}}^t\|^2$$

where the second inequality is because $\sum_{i=1}^{N} \rho_i^{t+1}(F_i(\bar{\mathbf{W}}^t) - F_i(\mathbf{W}_i^*)) \geq 0$ and $\eta_t L - 1 \leq -\frac{3}{4}$, and the last inequality is because $\frac{\gamma_t}{2\eta_t} \leq 1$ and $4\eta_t^3 L^2\Omega \geq 0$.

Plugging in everything into $A_1$, we can bound the effect of one-step SGD as

$$\|\bar{\mathbf{W}}^{t+1} - \mathbf{W}^*\|^2 \leq (1 - \eta_t\mu)\|\bar{\mathbf{W}}^t - \mathbf{W}^*\|^2 + 2\sum_{i=1}^{N} \rho_i^{t+1}\left(\|\bar{\mathbf{W}}^t - \mathbf{W}_i^t\|^2\right) \tag{22}$$

$$+ 6L\eta_t^2\Omega + 2\eta_t(\Gamma - \Omega) + \eta_t^2\|\bar{\mathbf{g}}_t - \mathbf{g}_t\|^2.$$

*Proof of Lemma 2.*

Assume Assumption 3 hold, the variance of gradients on all devices is bounded $\mathbb{E}\|\nabla f_i(\mathbf{W}^t;\xi_i) - \nabla F_i(\mathbf{W}^t)\|^2 \leq \sigma_i^2 \quad \forall i \in \{1,...,N\}$.

$$\mathbb{E}\|\mathbf{g}_t - \bar{\mathbf{g}}_t\|^2 = \mathbb{E}\left[ \left\| \sum_{i=1}^{N} \rho_i^{t+1}\nabla f_i(\mathbf{W}^t;\xi_i) - \sum_{i=1}^{N} \rho_i^{t+1}F_i(\mathbf{W}^t) \right\|^2 \right] \tag{23}$$

$$\leq \sum_{i=1}^{N}(\rho_i^{t+1})^2\mathbb{E}\left[\left\|\nabla f_i(\mathbf{W}^t;\xi_i) - F_i(\mathbf{W}^t)\right\|^2\right]$$

$$\leq \sum_{i=1}^{N}(\rho_i^{t+1})^2\sigma_i^2 \leq \max_t \sum_{i=1}^{N}(\rho_i^{t+1})^2\sigma_i^2$$

where the first inequality comes from the convexity of norms and Jensen's inequality.

*Proof of Lemma 3.*

Assume Assumption 4 holds, i.e., $\mathbb{E}\|\nabla f_i(\mathbf{W}^t; \xi_i)\|^2 \leq G^2 \quad \forall i \in \{1, ..., N\}$. Let $t_0$ denote a synchronization step. This means that $\mathbf{W}_i^{t_0} = \bar{\mathbf{W}}^{t_0}$. Because FL requires synchronization of every $E$ step, we have that $t - t_0 \leq E - 1$ where $t$ is any step between now and the next synchronization step (inclusively). Furthermore, we assume the learning rate $\eta_t$ is non-increasing and $\eta_o \leq 2\eta_t$. Then,

$$
\begin{aligned}
\mathbb{E}\left[\sum_{i=1}^{N} \rho_i^{t+1}\|\bar{\mathbf{W}}^t - \mathbf{W}_i^t\|^2\right] &= \mathbb{E}\left[\sum_{i=1}^{N} \rho_i^{t+1}\|(\bar{\mathbf{W}}^t - \bar{\mathbf{W}}^{t_0}) - (\mathbf{W}_i^t - \bar{\mathbf{W}}^{t_0})\|^2\right] \qquad (24) \\
&= \mathbb{E}\left[\mathbb{E}_{\rho^t}\|\mathbb{E}_{\rho^t}\left[(\mathbf{W}_i^t - \bar{\mathbf{W}}^{t_0})\right] - (\mathbf{W}_i^t - \bar{\mathbf{W}}^{t_0})\|^2\right] \\
&\leq \mathbb{E}\left[\mathbb{E}_{\rho^t}\|\mathbf{W}_i^t - \bar{\mathbf{W}}^{t_0}\|^2\right] \\
&= \mathbb{E}\left[\mathbb{E}_{\rho^t}\left\|\bar{\mathbf{W}}^{t_0} - \left(\bar{\mathbf{W}}^{t_0} - \sum_{i=0}^{t-1}\eta_t \nabla f_i(\mathbf{W}^t; \xi_i)\right)\right\|^2\right] \\
&= \mathbb{E}_{\rho^t}\left[\mathbb{E}\left\|\sum_{i=0}^{t-1}\eta_t \nabla f_i(\mathbf{W}^t; \xi_i)\right\|^2\right] \leq \mathbb{E}_{\rho^t}\left[\mathbb{E}\left\|\eta_0 \sum_{i=0}^{t-1} \nabla f_i(\mathbf{W}^t; \xi_i)\right\|^2\right] \\
&\leq \mathbb{E}_{\rho^t}\left[\mathbb{E}\left\|\eta_0 \sum_{i=0}^{t-1} \nabla f_i(\mathbf{W}^t; \xi_i)\right\|^2\right] \leq \mathbb{E}_{\rho^t}\left[\mathbb{E}\left[\eta_0^2(t - t_0)\left\|\nabla f_i(\mathbf{W}^t; \xi_i)\right\|^2\right]\right] \\
&\leq \mathbb{E}_{\rho^t}\left[\eta_0^2(E-1)G^2\right] \\
&\leq 4\eta_t^2(E-1)G^2
\end{aligned}
$$

### A.11 PROOF OF COROLLARY 1.1

Assume $\frac{N|\Xi_i|}{\sum_{j=1}^{N}|\Xi_j|}(F_i(\bar{\mathbf{W}}^t) - F_i(\mathbf{W}_i^*))$ are arranged in decreasing order, i.e., $\frac{N|\Xi_i|}{\sum_{j=1}^{N}|\Xi_j|}(F_i(\bar{\mathbf{W}}^t) - F_i(\mathbf{W}_i^*)) \geq \frac{N|\Xi_{i+1}|}{\sum_{j=1}^{N}|\Xi_j|}(F_{i+1}(\bar{\mathbf{W}}^t) - F_{i+1}(\mathbf{W}_{i+1}^*))$. If we choose $\mathcal{U}(*) \geq 0$ as a non-decreasing function, then it follows that,

$$
\begin{aligned}
&\frac{N|\Xi_i|}{\sum_{j=1}^{N}|\Xi_j|}(F_i(\bar{\mathbf{W}}^t) - F_i(\mathbf{W}_i^*)) \geq \frac{N|\Xi_{i+1}|}{\sum_{j=1}^{N}|\Xi_j|}(F_{i+1}(\bar{\mathbf{W}}^t) - F_{i+1}(\mathbf{W}_{i+1}^*)) \\
&\to \frac{N|\Xi_i|}{\sum_{j=1}^{N}|\Xi_j|}(F_i(\mathbf{W}_i^*) - F_i(\bar{\mathbf{W}}^t)) \leq \frac{N|\Xi_{i+1}|}{\sum_{j=1}^{N}|\Xi_j|}(F_{i+1}(\mathbf{W}_{i+1}^*) - F_{i+1}(\bar{\mathbf{W}}^t)) \\
&\to \mathcal{U}\left(\frac{N|\Xi_i|}{\sum_{j=1}^{N}|\Xi_j|}(F_i(\mathbf{W}_i^*) - F_i(\bar{\mathbf{W}}^t))\right) \leq \mathcal{U}\left(\frac{N|\Xi_{i+1}|}{\sum_{j=1}^{N}|\Xi_j|}(F_{i+1}(\mathbf{W}_{i+1}^*) - F_{i+1}(\bar{\mathbf{W}}^t))\right)
\end{aligned}
$$

Let's define $\tilde{\rho}_i^{t+1} \propto \mathcal{U}\left(\frac{N|\Xi_i|}{\sum_{j=1}^{N}|\Xi_j|}(F_i(\mathbf{W}_i^*) - F_i(\bar{\mathbf{W}}^t))\right)$, we have

$$
\sum_{i=1}^{N} \tilde{\rho}_i^{t+1} \frac{N|\Xi_i|}{\sum_{j=1}^{N}|\Xi_j|}(F_i(\bar{\mathbf{W}}^t) - F_i(\mathbf{W}_i^*)) \leq \frac{1}{N}\sum_{i=1}^{N}\frac{N|\Xi_i|}{\sum_{j=1}^{N}|\Xi_j|}(F_i(\bar{\mathbf{W}}^t) - F_i(\mathbf{W}_i^*))
$$

where the inequality is a direct consequence of the Chebyshev's sum inequality.

Then rewrite the equation above as the following,

$$
\sum_{i=1}^{N} \rho_i^{t+1}(F_i(\bar{\mathbf{W}}^t) - F_i(\mathbf{W}_i^*)) \leq \sum_{i=1}^{N} \rho_i(F_i(\bar{\mathbf{W}}^t) - F_i(\mathbf{W}_i^*))
$$

where $\rho_i^{t+1} = \frac{N|\Xi_i|}{\sum_{j=1}^{N}|\Xi_j|}\tilde{\rho}_i^{t+1}$ and $\rho_i = \frac{|\Xi_i|}{\sum_{j=1}^{N}|\Xi_j|}$.

Therefore,

$$\min_t \sum_{i=1}^{N} \rho_i^{t+1}(F_i(\bar{\mathbf{W}}^t) - F_i(\mathbf{W}_i^*)) \le \min_t \sum_{i=1}^{N} \rho_i(F_i(\bar{\mathbf{W}}^t) - F_i(\mathbf{W}_i^*)).$$

### A.12 EXTENSION TO PARTIAL PARTICIPATION

To provide an partial participation extension to Thm. 1, we need to define some additional notations. Note that the following derivation and notations largely follow the prior work (Li et al., 2019), which provides an easy way to extend FL convergence analysis to the partial participation setting.

**Stochasticity due to Client Sampling.** Now, instead of full participation of $N$ clients, at time $t$, we assume to sample $K$ clients from the pool of $N$ available clients, forming an active set of $\mathcal{S}^{t+1}$. This new *client sampling* procedure introduces another level of stochasticity in addition to *data sampling* stochasticity on each client. We use the notation $\mathbb{E}_s[\cdot]$ and $\mathbb{E}[\cdot]$ to denote expectation w.r.t each of the stochasticity respectively.

**Assumption 5** *The active set $\mathcal{S}^{t+1}$ is constructed by sampling a client with probabilities $\{\rho_i^{t+1}|i = 1, ..., N\}$ repeatedly for $K$ times with replacement, and the aggregation pattern is defined as,*

$$\tilde{\mathbf{W}}^{t+1} = \sum_{i=1}^{N} \rho_i^{t+1} \mathbf{W}_i^{t+1} \quad where \quad \mathbf{W}_i^{t+1} = \left\{ \begin{array}{ll} \mathbf{V}_i^{t+1}, & \text{if } t+1 \notin \mathcal{I}_E \\ \frac{1}{K} \sum_{i=1}^{K} \mathbf{V}_i^{t+1}, & \text{if } t+1 \in \mathcal{I}_E \end{array} \right\}. \quad (25)$$

The virtual sequence $\tilde{\mathbf{W}}^{t+1}$ is different than the virtual sequence $\bar{\mathbf{W}}^{t+1}$ in Eq. 2 when $t + 1 \in \mathcal{I}_E$. Therefore, the key to incorporate partial participation is characterizing the difference between the two when $t + 1 \in \mathcal{I}_E$.

To facilitate the proof we present the following two lemmas.

**Lemma 4** *If $t + 1 \in \mathcal{I}_E$, then*

$$\mathbb{E}_s[\tilde{\mathbf{W}}^{t+1}] = \bar{\mathbf{W}}^{t+1}.$$

**Lemma 5** *If $t + 1 \in \mathcal{I}_E$ and $\eta_t \le 2\eta_{t+E}$ is non-increasing $\forall t \ge 0$, then*

$$\mathbb{E}_s[\|\tilde{\mathbf{W}}^{t+1} - \bar{\mathbf{W}}^{t+1}\|^2] \le \frac{4}{K} \eta_t^2 E G^2$$

**Theorem 2** *Assume Assumptions 1-5 hold and $L, \mu, \sigma_i, G$ be defined therein. Choose $\gamma = \max \frac{L}{\mu}$ and the learning rate $\eta_t = \frac{2}{\mu(\gamma+t)}$ and $T \in \mathcal{I}_E$. Then FedAvg using SGD with partial device participation and a generic, time-varying sampling weights $\sum_{i=1}^{N} p_i^t = 1$ satisfies*

$$\mathbb{E}[F(\mathbf{W}^T)] - F(\mathbf{W}^*) \le \underbrace{\frac{L}{(\gamma+T)} \left( \frac{2(\bar{B}+C)}{\mu^2} + \frac{\gamma}{2}\Delta_0 \right)}_{vanishing} + \underbrace{\frac{L}{\mu}(\Gamma - \Omega)}_{non-vanishing}$$

*where $\Delta_0 = \|\mathbf{W}^0 - \mathbf{W}^*\|_2^2$, $\bar{B} = \max_t \sum_{i=1}^{N} (p_i^{t+1})^2 \sigma_i^2 + 8(E-1)G^2 + 6L\Omega$, $C = \frac{4}{K} E G^2$, $\Gamma = \max_t \sum_{i=1}^{N} \rho_i^{t+1}(F_i(\mathbf{W}^*) - F_i(\mathbf{W}_i^*))$, and $\Omega = \min_t \sum_{i=1}^{N} \rho_i^{t+1}(F_i(\bar{\mathbf{W}}^t) - F_i(\mathbf{W}_i^*)) \forall t \ge 0$.*

*Proof of Lemma 4*

$$\mathbb{E}_s[\tilde{\mathbf{W}}^{t+1}] = \mathbb{E}_s \left[ \frac{1}{K} \sum_{i=1}^{K} \mathbf{V}_i^{t+1} \right] = \frac{1}{K} \sum_{i=1}^{K} \mathbb{E}_s \left[ \mathbf{V}_i^{t+1} \right] = \sum_{i=1}^{N} \rho_i^{t+1} \mathbf{V}_i^{t+1} = \bar{\mathbf{W}}^{t+1}$$

*Proof of Lemma 5*

$$\mathbb{E}_s[\|\tilde{\mathbf{W}}^{t+1} - \bar{\mathbf{W}}^{t+1}\|^2] = \mathbb{E}_s\left[\left\|\frac{1}{K}\sum_{i=1}^{K}\mathbf{V}_i^{t+1} - \bar{\mathbf{W}}^{t+1}\right\|^2\right] = \mathbb{E}_s\left[\frac{1}{K^2}\left\|\sum_{i=1}^{K}(\mathbf{V}_i^{t+1} - \bar{\mathbf{W}}^{t+1})\right\|^2\right]$$

$$\leq \mathbb{E}_s\left[\frac{1}{K^2}\sum_{i=1}^{K}\left\|\mathbf{V}_i^{t+1} - \bar{\mathbf{W}}^{t+1}\right\|^2\right] = \frac{1}{K^2}\sum_{i=1}^{K}\mathbb{E}_s\left[\left\|\mathbf{V}_i^{t+1} - \bar{\mathbf{W}}^{t+1}\right\|^2\right]$$

$$= \frac{1}{K}\mathbb{E}_s\left[\left\|\mathbf{V}_i^{t+1} - \bar{\mathbf{W}}^{t+1}\right\|^2\right]$$

where the first inequality stems from triangle inequality of norms. Now we introduce a new notation $t_s \doteq t + 1 - E \in \mathcal{I}_E$, which is the most recent aggression moment. Therefore, $\bar{\mathbf{W}}^{t_s}$ is the same across all clients.

$$\frac{1}{K}\mathbb{E}_s\left[\left\|\mathbf{V}_i^{t+1} - \bar{\mathbf{W}}^{t_s} + \bar{\mathbf{W}}^{t_s} - \bar{\mathbf{W}}^{t+1}\right\|^2\right] = \frac{1}{K}\mathbb{E}_s\left[\left\|(\mathbf{V}_i^{t+1} - \bar{\mathbf{W}}^{t_s}) - (\bar{\mathbf{W}}^{t+1} - \bar{\mathbf{W}}^{t_s})\right\|^2\right]$$

$$= \frac{1}{K}\mathbb{E}_s\left[\left\|(\mathbf{V}_i^{t+1} - \bar{\mathbf{W}}^{t_s}) - \mathbb{E}_s[\mathbf{V}_i^{t+1} - \bar{\mathbf{W}}^{t_s}]\right\|^2\right] \leq \frac{1}{K}\mathbb{E}_s\left[\left\|\mathbf{V}_i^{t+1} - \bar{\mathbf{W}}^{t_s}\right\|^2\right]$$

The last inequality stems from the calculation of auto-correlation, i.e., $\mathbb{E}[\|x - \mathbb{E}[x]\|^2] = \mathbb{E}\|x\|^2 - \mathbb{E}[x]^2$. Finally, we have the following,

$$\mathbb{E}\left[\mathbb{E}_s[\|\tilde{\mathbf{W}}^{t+1} - \bar{\mathbf{W}}^{t+1}\|^2]\right] \leq \frac{1}{K}\sum_{i=1}^{N}\rho_i^{t+1}\left(\mathbb{E}\left[\left\|\mathbf{V}_i^{t+1} - \bar{\mathbf{W}}^{t_s}\right\|^2\right]\right)$$

$$= \frac{1}{K}\sum_{i=1}^{N}\rho_i^{t+1}\left(\mathbb{E}\left[\left\|\sum_{j=t_s}^{t}\eta_j\nabla f_i(\mathbf{W}_i^j;\xi_i^j)\right\|^2\right]\right)$$

$$\leq \frac{1}{K}\sum_{i=1}^{N}\rho_i^{t+1}\left(\sum_{j=t_s}^{t}\mathbb{E}\left[\left\|\eta_j\nabla f_i(\mathbf{W}_i^j;\xi_i^j)\right\|^2\right]\right)$$

$$\leq \frac{1}{K}\sum_{i=1}^{N}\rho_i^{t+1}\left(4\eta_t^2\sum_{j=t_s}^{t}\mathbb{E}\left[\left\|\nabla f_i(\mathbf{W}_i^j;\xi_i^j)\right\|^2\right]\right)$$

$$\leq \frac{1}{K}\sum_{i=1}^{N}\rho_i^{t+1}\left(4\eta_t^2 EG^2\right) = \frac{4}{K}\left(\eta_t^2 EG^2\right)$$

## A.13 ADDITIONAL PROOF

In the main paper, we claimed equality between Thm. 1 and the convergence bound in a prior work (Li et al., 2019) if $\rho_i^t = \rho_i = \frac{|\Xi_i|}{\sum_{j=1}^{N}|\Xi_j|}$. Specifically, we want to show that $\Omega = \Gamma = \sum_{i=1}^{N}\rho_i(F_i(\mathbf{W}^*) - F_i(\mathbf{W}_i^*))$. In this section, we give a detailed proof to this statement. The equality holds because

$$\Omega = \min_t \sum_{i=1}^{N}\rho_i(F_i(\bar{\mathbf{W}}^t) - F_i(\mathbf{W}_i^*)) = \min_t\left[\sum_{i=1}^{N}\rho_i F_i(\bar{\mathbf{W}}^t)\right] - \sum_{i=1}^{N}\rho_i F_i(\mathbf{W}_i^*)$$

$$= \sum_{i=1}^{N}\rho_i F_i(\mathbf{W}^*) - \sum_{i=1}^{N}\rho_i F_i(\mathbf{W}_i^*),$$

and,

$$\Gamma = \max_t \sum_{i=1}^{N}\rho_i(F_i(\mathbf{W}^*) - F_i(\mathbf{W}_i^*)) = \sum_{i=1}^{N}\rho_i F_i(\mathbf{W}^*) - \sum_{i=1}^{N}\rho_i F_i(\mathbf{W}_i^*).$$

Therefore, the bound in Thm 1 reduces to the following,

$$\mathbb{E}[F(\mathbf{W}^T)] - F(\mathbf{W}^*) \leq \underbrace{\frac{L}{(\gamma + T)} \left( \frac{2\bar{B}}{\mu^2} + \frac{\gamma}{2}\Delta_0 \right)}_{vanishing}$$

where $\Delta_0 = \|\mathbf{W}^0 - \mathbf{W}^*\|_2^2$, $\bar{B} = \sum_{i=1}^N \rho_i^2 \sigma_i^2 + 8(E-1)G^2 + 6L\Gamma$, $\Gamma = \sum_{i=1}^N \rho_i F_i(\mathbf{W}^*) - \sum_{i=1}^N \rho_i F_i(\mathbf{W}_i^*)$.

