# OpenReview forum: "Exp-$\alpha$: Beyond Proportional Aggregation in Federated Learning"
_ICLR.cc/2023/Conference — Submitted to ICLR 2023_

### Official Review · Reviewer_gbV4 · 2022-10-23

**Confidence:** 3
**Correctness:** 3
**Technical Novelty And Significance:** 3
**Empirical Novelty And Significance:** 3
**Recommendation:** 5

**Clarity, Quality, Novelty And Reproducibility:**

The paper is well written, and is novel to my knowledge, reproducibility is unclear to me.

**Strength And Weaknesses:**

Strengths:
- The paper is well-motivated and well-writtem, the story line and the theoretical analysis are easy to follow.
- Experiments are thorough.

Weakness:
- The theory developed (Theorem 1) is not very strong. The definition of $\Gamma$ and $\Omega$ actually depends on $T$, so it is more accurate to write them as $\Gamma_T$ $\Omega_T$. As a result, the convergence error (the non-vanishing bias term) depends on $T$, which could potentially goes to 0 as $T \to \infity$. The relationship between the bias term and $T$ making the trade-off between convergence rate and convergence error less clear. Please also see my questions below.

Questions:
1. Theorem 1 assumes full participation, is it possible to extend it to the parital participation case? This seems to be important because the full participation assumption is too strong for FL.
2. Consider the proportional aggregation strategy, the bound in Theorem 1 reduces to
$$
E[ F(W^t) ] - F(W^*) \leq \frac{L}{\gamma + T} O(1) + \frac{L}{\mu} \min_{t \in [T]} \sum_{i=1}^N \rho_i ( F_i( W^t ) - F_i(W^*) ) \\
\leq \frac{L}{\gamma + T} O(1) + \frac{L}{\mu} \min_{t \in [T]} \sum_{i=1}^N ( F(W^t) - F(W^*) )
$$
which is trivial since $L/\mu > 1$. This sanity check indicates that the inequality is rather loose. Is it possible to tighten the bound so that we can get a non-trivial bound for proportional aggregation?




**Summary Of The Paper:**

The author(s) study the model aggregation strategy in federated learning. The author(s) conduct a theoretial analysis on the impact of the model aggregation strategy. In particular, the author(s) show that the model aggregation can lead to a trade-off between convergence rate and convergence error (a non-vanishing bias term). Experiments on real-world datasets under various settings are conducted to support the proposed method.

**Summary Of The Review:**

The paper is well-written and the experiments are thorough and clear to me. My only concern is on the theory side, please see my questions in the "Strength And Weaknesses" section. I will raise my score if these questions can be addressed.

---

> ### Author Response · Authors · 2022-11-15
> **Added extension to partial participation and explained that our bound is non-trivial**
>
> We appreciate the reviewer’s positive comments on experiments and suggestions to include partial participation. We have added new theoretical results and discussion to clarify the reviewer’s concerns.
>
> **1, Theory 1 not strong. Definition of $\Gamma$ and $\Omega$ depends on time. So the non-vanishing term could go to zero.**
>
> $\Gamma$ and $\Omega$ are *independent* of time (T) because of the max and min over time operators (see the definitions below). For example, $c=\max_t f(t), \forall t\geq0$ is a constant not depending on time. The non-vanishing term **does not** go to zero unless proportional aggregation is used. Specifically, the non-vanishing term is the difference between two terms, $\Gamma$ and $\Omega$. Therefore, the non-vanishing term will only become zero when they are equal. We have shown that it becomes zero only when the aggression weights are the same as proportional aggregation (Appendix A.13).
> In Theorem 1, we defined $\Gamma$ and $\Omega$ as the following,
>
> $\Gamma = \max_t \sum_{i=1}^N \rho_i^{t+1} \left(F_i(\mathbf{W}^*) - F_i(\mathbf{W}^*_i) \right)$, and
>
>  $\Omega= \min_t \sum_{i=1}^N \rho_i^{t+1} (F_i(\bar{\mathbf{W}}^{t}) -F_i(\mathbf{W}^*_i)) \forall t\geq0.$
>
> **2, Extension to partial participation**
>
> We thank the reviewer for the motivation of partial participation. We have extended our theoretical analysis to the partial participation setting in Appendix A.12. Just as in Theorem 1, we still observe a trade-off between convergence rate and error in the new theorem. Intuitively, partial participation introduces a new level of stochasticity due to client sampling in addition to stochasticity due to data sampling. The added stochasticity manifests in a slower convergence in the form of an additional constant term in the non-vanishing component.
>
> **3, Non-trivial bound for proportional aggregation**
>
> We appreciate the reviewer’s efforts in checking our derivation in detail. Our bound is non-trivial for proportional aggregation as stated at the end of Sec.3.2  in the main paper. The reviewer claims that because $L/\mu >0$, the bound is rather loose. In fact,  $L/\mu (\Gamma-\Omega)$ (the non-vanishing term in Theorem 1) term is actually **zero** when using proportional aggregation despite $L/\mu >0$. This makes the bound tight. We have revised our proof to show this more clearly in Appendix A.13. Furthermore, when instantiated with proportional aggregation, the bound matches the bound in prior work [1]. This further shows that our bound for proportional aggregation is non-trivial. Specifically, you will see that the bound for proportional aggregation is the following,
>
> $ \mathbb{E}[F(\mathbf{W}^T)] - F(\mathbf{W}^*) \leq  {\frac{L}{(\gamma+T)}\left( \frac{2\bar{B}}{\mu^2} + \frac{\gamma}{2}\Delta_0\right)} $
>
> where  $\Delta_0=\|\mathbf{W}^0-\mathbf{W}^*\|^2_2$, $\bar{B} =  \sum_{i=1}^N \rho_i^2\sigma_i^2+ 8(E-1)G^2 + 6L\Gamma$ and
>
> $\Gamma = \sum_{i=1}^N \rho_i F_i(\mathbf{W}^*) - \sum_{i=1}^N \rho_iF_i(\mathbf{W}^*_i).$
>
> [1] Li, Xiang, et al. "On the convergence of fedavg on non-iid data." ICLR (2020).
>
> With the clarification and additional theoretical analysis, we hope that we have answered all your questions. Please let us know if there are more questions.

---

> > ### Author Response · Authors · 2022-11-25
> > **Thank you for your reviews!**
> >
> > Dear Reviewer gbv4,
> >
> > We would like to provide a short summary of our response to your questions to further help you read the rebuttal.
> >
> > * We clarified that $\Gamma$ and $\Omega$ are **time-independent** (Q1)
> > * We provided a **new theoretical extension to partial aggregation** (Q2)
> > * We showed that **our bound for proportional aggregation is non-trivial** (Q3)
> >
> > Thank you for the constructive comments and happy holidays!
> >
> > Authors of 3299

---

> > ### Comment · Reviewer_gbV4 · 2022-12-05
> > **Thanks for your response**
> >
> > I would like to thank the author(s) for the responses, which have address my major concern. I am raising my score.

---

> > ### Comment · Reviewer_gbV4 · 2022-12-10
> > **After on online discussion**
> >
> > After the online discussion with AC and other reviewers, I think Reviewer Cggb's question is very valid, I also encourage the author(s) to provide some in-depth theoretical analysis of the aggregation strategy proposed in Section 3.4. The Exp-\alpha aggregation strategy seems to be a practical heuristic in the current version of the manuscript. I am changing my rating from 6 to 5.

---

### Official Review · Reviewer_CKau · 2022-10-25

**Confidence:** 3
**Clarity, Quality, Novelty And Reproducibility:** See the detailed strength/weakness se…
**Correctness:** 3
**Technical Novelty And Significance:** 2
**Empirical Novelty And Significance:** 2
**Recommendation:** 6

**Strength And Weaknesses:**

Strengths
- The theoretical analysis using time-varying aggregation weights reveals that setting proportional weights is a suboptimal strategy, especially in terms of the convergence error.
- Thorough experiments across different datasets and with different types of heterogeneity.
- The proposed aggregation strategy, $Exp-\alpha$, is easy to apply in practice and  outperforms baseline proportional aggregation in all scenarios.

Weaknesses

- **Metrics** : I assume that the authors report test accuracy of the global model averaged across clients. This wasn't clarified in the Experiments section or the appendix. Additionally, it would be  interesting to see the performance of the global model only on the heterogeneous clients, i.e., the ones which $Exp-\alpha$ down-weights during aggregation. Further, robust algorithms perform well under low levels of corruption, so it would be interesting to see the performance of $Exp-\alpha$ when half of the clients are corrupted.

- **Theoretical results hold only for strongly convex cases** : Since the authors have not adapted proofs for convex or non-convex cases, the exact expression of proportional aggregation might be different in these cases.

- **Domain Shift Experiments**: From Table 3, I could see that $Exp-\alpha$ performs worse than proportional aggregation on several domains, however, the average accuracy is better. Could the authors provide an explanation for this? I expect that the aggregation weights for clients from these domains should be small but a further analysis of how heterogeneous these domains are, might help.

- **Effects of $\alpha$, heterogeneity and number of local steps** : Appendix A.5 is difficult to parse, especially Figures 3,4 and 5. (No legend in Figures 3 and 4 and Figure 5 is too small). While the authors do have all these sets of experiments, they do not compare how aggregation differs wrt increasing heterogeneity and number of local steps. I think that the value of $\alpha$ controls how different the proportional weights are, with lower $\alpha$ providing larger difference. Using this explanation, increasing heterogeneity among clients or increasing the number of local steps, should increase the deviation between local models and global models and should require a smaller $\alpha$ to handle it. Could the authors provide more details on this?


Suggestions
- **Explaining Experiments** : The set of experiments are exhaustive, however, some analysis of the experimental results in terms of behavior wrt number of local steps, the value of $\alpha$ and the cases where proportional aggregation outperforms $Exp-\alpha$ would greatly benefit the paper.

- **Possible connection to robust aggregation**: Note that several robust aggregation algorithms, for instance Trimmed Mean[1], discard or decrease the weight of clients which differ severely from the average. Here, $Exp-\alpha$ also gives lower weights to clients which differ a lot from the average iterate and should ideally also provide some "robustness" in the aggregation.
- **$Exp-\alpha$ for stateful algorithms**: Working out the proof using a baseline stateful algorithm, for instance SCAFFOLD[2], should provide more insights into whether stateful algorithms also suffer from the issue of suboptimal convergence rate with proportional aggregation or not.


### Typos --
- Theorem~1 : What is the max over? and doesn't it require Assumption 1 (Strong Convexity)
- Figures 3 and 4 in the appendix do not have legends so it is difficult to figure out exactly what is the effect of $\alpha$.


### References --

[1] Yin, D., Chen, Y., Kannan, R. &amp; Bartlett, P.. (2018). Byzantine-Robust Distributed Learning: Towards Optimal Statistical Rates. <i>Proceedings of the 35th International Conference on Machine Learning</i>, in <i>Proceedings of Machine Learning Research</i> 80:5650-5659 Available from https://proceedings.mlr.press/v80/yin18a.html.


[2] Karimireddy, S.P., Kale, S., Mohri, M., Reddi, S., Stich, S. &amp; Suresh, A.T.. (2020). SCAFFOLD: Stochastic Controlled Averaging for Federated Learning. <i>Proceedings of the 37th International Conference on Machine Learning</i>, in <i>Proceedings of Machine Learning Research</i> 119:5132-5143 Available from https://proceedings.mlr.press/v119/karimireddy20a.html.


**Summary Of The Paper:**

By analyzing the effect of time-varying aggregation weights in FL, the authors obtain a tradeoff between the convergence rate and the convergence error, where FedAvg favors the latter. Further, setting the aggregation weights proportional to the change in local loss after the local steps, the authors obtain a new aggregation strategy, $Exp-\alpha$. This algorithm outperforms proportional aggregation across several baseline FL algorithms on several FL datasets, in terms of both the convergence rate and the final convergence error.


**Summary Of The Review:**

See the detailed strength/weakness section

---

> ### Author Response · Authors · 2022-11-15
> **Explained metrics and added discussion with new experiments**
>
> We appreciate the reviewer's positive comments on the thoroughness of the experiments. We aim to go beyond theory to test the effectiveness of our algorithm. We would like to provide the following clarifications and discussion.
>
> **1, Metrics**
>
> We are sorry for the confusion. We follow prior works for reporting performance. Specifically, for the imbalance and label-flipping experiments, following [1], we report the performance of the global model on a standalone test set. For the domain-shift experiment, following [2], we report the performance of the global model (or local model when using FedBN [2]) *on each client*. Note that in this case, each client has access to data from a unique domain not shared by other clients (Sec. 4.3). For example, in Tab.3, the Digits benchmark consists of five domains. Therefore, there are five clients in total. So the domain-shift experiment is a good indicator of performance of the global model on the heterogeneous clients, which are down-weighted. More details are provided in response to questions 3 on domain-shift experiments.
>
> If we understand correctly the reviewer’s definition of “low level of corruption” as having half or lower number of clients being corrupted, we have already included experiments in the paper. In Fig 2.b (main paper), we experimented with different levels of corruption including very low levels of corruption (even lower than half). Furthermore, we also have experiments for partial corruption, i.e., only a fraction of labels are flipped on a corrupted client, a setting that is more practical then the settings in the paper [3] referenced by the reviewer. As shown in Fig 2.b, across four different levels of corruption from low to high and including parietal corruption, Exp-$\alpha$ produces highly intuitive aggregation patterns, adapting to the corruption levels of individual clients, and improves performance over conventional proportional aggregation consistently.
>
> [1] McMahan, Brendan, et al. "Communication-efficient learning of deep networks from decentralized data." Artificial intelligence and statistics. PMLR, 2017.
>
> [2] Li, Xiaoxiao, et al. "Fedbn: Federated learning on non-iid features via local batch normalization." arXiv preprint arXiv:2102.07623 (2021).
>
> [3] Yin, Dong, et al. "Byzantine-robust distributed learning: Towards optimal statistical rates." International Conference on Machine Learning. PMLR, 2018.
>
>
> **2, Theoretical results hold only for strongly convex cases**
>
> We agree with the reviewer’s comments. Nonetheless, note that we already relax assumptions in some existing works, namely fixed aggregation weights, and our analysis already serves a purpose to demonstrate the trade-off between speed and error and is a significant contribution.
>
> Extending the current analysis to other assumptions on top of our existing relaxation is difficult, especially because we already relaxed the assumption on fixed aggregation weights that most existing algorithms have [1-6]. Existing works implicitly assume that the aggregation weights are either proportional or uniform across clients whereas our analysis uses a time-varying aggregation weights. With this additional relaxation, we are only able to provide convergence bound under more restrictive FL assumptions following [1,2,3], e.g., strong convexity (Assumption 1) currently.
>
> Essentially, with the strong convex assumption, we are able to quantify the degree of heterogeneity among clients in the form of difference between the global and local risks, i.e,$\Omega$ and $\Gamma$ in Theorem 1, in which the role of aggregation weights becomes clear. For example, in Theorem 1, we defined $\Omega$ as,
>
> $ \Omega = \min_t \sum_{i=1}^N \rho_i^{t+1} (F_i(\bar{\mathbf{W}}^{t}) -F_i(\mathbf{W}^*_i))$
>
> In contrast, many other analyses rely on a *bounded similarity assumption* to bound the heterogeneity among clients as shown below.
>
> $\mathbb{E}\|\nabla F_i(\mathbf{W})-\nabla F(\mathbf{W})\|^2 \leq \xi^2 +\beta^2\|\nabla F(\mathbf{W})\|^2$
>
> where $\xi$ and $\beta$ are parameters. Namely, instead of capturing heterogeneity in the form of risk differential, they introduce bounded gradient assumption, which reduces the effect of heterogeneity to a constant [4,5,6]. This obscures the dependence of aggregation strategy on heterogeneity.
>
> Extending the current analysis to other assumptions is an ongoing work. We believe that elucidating this tradeoff with our time-varying aggregation analysis, and thoroughly-validated practical algorithm that results from this observation, is a strong contribution.
>
> [1] Li, Xiang, et al. "On the convergence of fedavg on non-iid data." ICLR (2020).
>
> [2] Deng, Yuyang, Mohammad Mahdi Kamani, and Mehrdad Mahdavi. "Distributionally robust federated averaging." NeurIPS (2020)
>
> [3] Cho, Yae Jee, Jianyu Wang, and Gauri Joshi. "Client selection in federated learning: Convergence analysis and power-of-choice selection strategies." AISTATS (2022).

---

> > ### Author Response · Authors · 2022-11-15
> > **Cont’d**
> >
> >
> > [4] Gorbunov, Eduard, Filip Hanzely, and Peter Richtárik. "Local sgd: Unified theory and new efficient methods." International Conference on Artificial Intelligence and Statistics. PMLR, 2021.
> >
> > [5] Karimireddy, Sai Praneeth, et al. "Mime: Mimicking centralized stochastic algorithms in federated learning." arXiv preprint arXiv:2008.03606 (2020).
> >
> > [6] Karimireddy, Sai Praneeth, et al. "Scaffold: Stochastic controlled averaging for federated learning." International Conference on Machine Learning. PMLR, 2020.
> >
> > **3, Domain Shift Experiments**
> >
> > Yes, the lower performance on some domains is due to lower assigned aggregation weights by Exp-$\alpha$, which is correlated with how heterogeneous the corresponding domains are w.r.t the overall data distribution. We can obtain more insights by delving deeper into existing experiments. We will use the Digits benchmark in Tab.3 (main paper) as an example because the difference is the most noticeable on this benchmark. The Digits benchmark consists of five datasets.
> >
> > - MNIST: black-and-white single digits
> >
> > - SVHN: house numbers from street view
> >
> > - USPS: scanned handwritten digits from U.S Postal Service
> >
> > - Synthetic Digits (SD): synthetically generated digits with noisy backgrounds.
> >
> > - MNIST-M: MNIST digits with random background from colored photos
> >
> > In the first row of Tab.3, where FedAvg with promotional aggregation is used, we can rank the performance of a global model on each of the dataset as the following:
> >
> > MNIST (95.64) > USPS (95.38) > SD (81.82) > MNIST-M (75.26) > SVHN (61.33).
> >
> > From this result, we expect that MNIST, USPS and SD are more similar to each other than MNIST-M and SVHN are. In fact, the prior works [1] confirms the same finding by plotting a histogram of pixel values from each of the datasets. Now, if we look closer at the results of FedBN with proportional aggregation (Propoto.) and with Exp-$\alpha$ in Tab.3., Exp-alpha outperforms Propoto on MNIST, USPS and SD. which happen to be the more similar datasets. In other words, Exp-$\alpha$ gives lower weights to domains with more heterogeneity w.r.t the overall data distribution. This behavior generally yields better overall (averaged) performance across all clients as pointed out by the reviewer. While there are some fluctuations on other more realistic benchmarks (Office-Caltech10 and DomainNet in Tab.3), the differences between Propoto. and Exp-$\alpha$ are not nearly as drastic as on the digits benchmark. This suggests that real-world datasets more likely suffer from only mild domain shifts.
> >
> > [1] Peng, Xingchao, et al. "Moment matching for multi-source domain adaptation." Proceedings of the IEEE/CVF international conference on computer vision. 2019.
> >
> > **4, Effects of $\alpha$  heterogeneity and number of local steps**
> >
> > Thank you for bringing this up and we have fixed the figures in Appendix. Yes, your intuition is correct. With increasing heterogeneity among clients or increasing number of local steps, a smaller $\alpha$ can do better. To demonstrate this we add the following experiments. Specifically, we use the imbalance benchmark as an example. At each round of global communication, we sample a different set of 10 clients, six of which are imbalanced.
> >
> > To show the effects of increasing local steps, we run experiments across four communication settings with an increasing number of local steps: E = {20,40,80,160} and a fixed imbalance ratio of 0.1. For each setting, we sweep $\alpha \in$ {0.2,1,5,25,125}. A total of 20 experiments are conducted. In the following table, we show the best $\alpha$ for each communication setting. This confirms the reviewer’s intuition that increasing number of local steps requires smaller $\alpha$;
> >
> > | E        | 20  | 40 | 80 | 160 |
> > |----------|-----|----|----|-----|
> > | $\alpha$ | 125 | 5  | 5  | 0.2 |
> >
> > To show the effects of increasing heterogeneity, we fix the number of local steps to be E=160 and vary the imbalance ratio in {0.1,0.2,0.3,0.4,0.5} with *smaller* numbers indicating *more severe* imbalance. Similarly, we sweep $\alpha \in$  {0.2,1,5,25,125}. An additional 20 experiments are conducted. In the following table we show the best $\alpha$ for each imbalance ratio. The result is consistent with the reviewer’s intuition that more severe heterogeneity can benefit from smaller $\alpha$.
> >
> > | Imbalance  | 0.1 | 0.2 | 0.3 | 0.4 | 0.5 |
> > |------------|-----|-----|-----|-----|-----|
> > | alpha      | 0.2 | 0.2 | 0.2 | 5   | 125 |
> >
> > We have added the discussion in appendix A5.

---

> > > ### Author Response · Authors · 2022-11-15
> > > **Cont’d**
> > >
> > > **5, Comparison to Robust algorithms**
> > >
> > > Robust FL algorithms such as [1], are specialized algorithms targeting Byzantine failures, e.g., random label flipping and adversarial attacks. They aim to provide algorithms and convergence guarantees under Byzantine failures. In contrast, our approach follows a more general derivation emphasizing the trade-off between convergence rate and error with no specific assumptions on Byzantine failures. Nonetheless, empirically, our algorithm shows very good performance against random label flipping under different levels of corruptions as already shown in Tab.2 (main paper), even though our analysis and experiments are more general. It would be interesting to theoretically study the exp-$\alpha$ under the scheme of Byzantine failures and compare to robust algorithms in more elaborated experimental settings.
> > >
> > > [1] Yin, Dong, et al. "Byzantine-robust distributed learning: Towards optimal statistical rates." International Conference on Machine Learning. PMLR, 2018.
> > >
> > > **6, Extension to SCAFFOLD**
> > >
> > > Extension to SCAFFOLD [1] is technically challenging due to additional assumptions in its convergence analysis. We carefully looked into the convergence analysis of SCAFFOLD. However, as mentioned earlier in response to Question 2, SCAFFOLD introduces a new assumption, *the bounded similarity assumption*, on the heterogeneity of clients. This assumption effectively reduces the effect of heterogeneity to a constant in the convergence bound. With this assumption, the dependence of aggregation strategy on heterogeneity becomes tangled. Removing this assumption will require significant deviation from its original analysis, which is not immediately clear.  Nonetheless, we are still actively looking into alternatives such that we can remove both the bounded similarity assumption and the fixed aggregation assumption in SCAFFOLD.
> > >
> > > [1] Karimireddy, Sai Praneeth, et al. "Scaffold: Stochastic controlled averaging for federated learning." International Conference on Machine Learning. PMLR, 2020.

---

> > > > ### Author Response · Authors · 2022-11-25
> > > > **Thank you for your reviews!**
> > > >
> > > > Dear Reviewer Ckau,
> > > >
> > > > We would like to provide a short summary of our response to your questions to further help you read the rebuttal.
> > > >
> > > > * We clarified the **definitions of metrics** (Q1).
> > > > * We pointed to experiments showing the **performance of the global model on heterogeneous clients** (Q1).
> > > > * We pointed to experiments showing **performance on low levels of corruption** (Q1).
> > > > * We discussed reasons **why relaxing strong convexity is difficult** (Q2).
> > > > * We discussed with experiments that aggregation weights for certain domains are indeed smaller which **explains why Exp-alpha performs worse on some domains** (Q3).
> > > > * We added **40+ new sets of experiments to analyze how $\alpha$ changes with increasing local steps and increasing heterogeneity** (Q4).
> > > > * We discussed **comparisons to robust algorithms** (Q5).
> > > > * We discussed the technical **difficulty of applying the current analysis to SCAFFOLD** (Q6).
> > > >
> > > > Thank you for the constructive comments and happy holidays!
> > > >
> > > > Authors of 3299

---

### Official Review · Reviewer_Cggb · 2022-10-30

**Confidence:** 3
**Correctness:** 2
**Technical Novelty And Significance:** 3
**Empirical Novelty And Significance:** 2
**Recommendation:** 5

**Clarity, Quality, Novelty And Reproducibility:**

This paper is well-written and easy to follow. The novelty of this paper is good, although the proposed time-varying aggregation and the Exp-$\alpha$ scheme are only limited to restrictive assumptions. The reproducibility of this paper seems to be good.

**Strength And Weaknesses:**

Strengths:
1. This paper considered model aggregation, which is an important aspect of FL.
2. The revealed insights based on a more general time-varying aggregation scheme are interesting.

Weaknesses:
1. Although this paper provides a fresh perspective on model aggregation in FL, the strong convexity (Assumption 1) and bounded local gradients (Assumption 4) are quite restrictive and not very interesting in the state-of-the-art convergence analysis of FL.

2. The main theoretical results in Theorem 1 have not been presented clearly. Specifically, $\Gamma$ and $\Omega$, the two most important quantities in Theorem 1, are quite confusing. For a while, it appears to the reviewer that they are asymptotic and independent of $t$. But later in Sec. 3.3, it seems that they are only for finite time $t$. I suggest changing the notations to $\Gamma_t$ and $\Omega_t$ to avoid such confusion. Also, the authors seem to have an incorrect understanding of the terminology "convergence rate," which typically characterizes the finite-time error of some convergence metric (gap w.r.t. the optimal value of the problem in Theorem 1) w.r.t. $T$. In Theorem 1, it is unclear how $\Gamma-\Omega$ scales w.r.t. $t$ when the proposed time-varying aggregation scheme is used. On the other hand, $\Gamma-\Omega = 0$ when the conventional proportional aggregation scheme is used. Thus, it is inaccurate to claim that the time-varying aggregation can achieve an improved convergence rate. In fact, the time-varying scheme may even hurt the convergence rate performance since $\Gamma_t - \Omega_t$ could have a convergence rate worse than $O(1/T)$.

3. For the proposed Exp-$\alpha$ aggregation scheme, the authors also didn't provide any theoretical analysis on its convergence rate in terms of $\Gamma-\Omega$. Thus, it is hard to know theoretically how fast the convergence rate performance of the Exp-$\alpha$ scheme is.

**Summary Of The Paper:**

This paper considered the impacts of model aggregation in federated learning (FL). The authors first derive the convergence performance of FL with a more general time-varying aggregation scheme for FL. Then by specializing the time-varying convergence result to the conventional proportional aggregation, the authors argued that the conventional proportional aggregation has its limitation in favoring convergence error while sacrificing convergence rate performance. Then, the authors proposed an exponential-type aggregation scheme called Exp-$\alpha$ and empirically showed that Exp-$\alpha$ achieves good performance.

**Summary Of The Review:**

This paper considered the impacts of model aggregation in FL and proposed a time-varying model aggregation scheme to improve the convergence of FL over the conventional proportional scheme. Although the paper reveals some interesting insights for the proposed time-varying scheme and empirically shows the performance of the Exp-$\alpha$ scheme, there are some confusions and potential misunderstandings of convergence rate in the theoretical results in this paper.

---

> ### Author Response · Authors · 2022-11-15
> **Clarified important confusion on time-dependence and the definition of convergence rate**
>
> **1, Strong convexity (Assumption 1) and bounded local gradients (Assumption 4)**
>
> We thank the reviewer for bringing up the discussion. Nonetheless, note that we already relax assumptions in some existing works, namely fixed aggregation weights, and our analysis already serves a purpose to demonstrate the trade-off between speed and error and is a significant contribution.
>
> Our analysis is novel because we relaxed the assumption on fixed aggregation weights that most existing algorithms have [1-6]. They assume that the aggregation weights are either proportional or uniform across clients whereas our analysis uses a time-varying aggregation weights. With this additional relaxation, we are only able to provide convergence bound under more restrictive FL assumptions following [1,2,3], e.g., strong convexity (Assumption 1) and bounded local gradient (Assumption 4) currently. Nonetheless, they serve a purpose to demonstrate the trade-off between speed and error.
>
> Essentially, with the strong convex assumption, we are able to quantify the degree of heterogeneity among clients in the form of difference between the global and local risks, i.e,$\Omega$ and $\Gamma$ in Theorem 1, in which the role of aggregation weights becomes clear. For example, in Theorem 1, we defined $\Omega$ as,
>
> $ \Omega = \min_t \sum_{i=1}^N \rho_i^{t+1} (F_i(\bar{\mathbf{W}}^{t}) -F_i(\mathbf{W}^*_i))$
>
> In contrast, many other analyses rely on a *bounded similarity assumption* to bound the heterogeneity among clients as shown below.
>
> $\mathbb{E}\|\nabla F_i(\mathbf{W})-\nabla F(\mathbf{W})\|^2 \leq \xi^2 +\beta^2\|\nabla F(\mathbf{W})\|^2$
>
> where $\xi$ and $\beta$ are parameters. Namely, instead of capturing heterogeneity in the form of risk differential, they introduce bounded gradient assumption, which reduces the effect of heterogeneity to a constant [4,5,6]. This obscures the dependence of aggregation strategy on heterogeneity.
>
> Relaxing the current analysis to other assumptions is a challenging and ongoing work. We believe that elucidating this tradeoff with our time-varying aggregation analysis, and thoroughly-validated practical algorithm that results from this observation, is a strong contribution.
>
> [1] Li, Xiang, et al. "On the convergence of fedavg on non-iid data." ICLR (2020).
>
> [2] Deng, Yuyang, Mohammad Mahdi Kamani, and Mehrdad Mahdavi. "Distributionally robust federated averaging." NeurIPS (2020)
>
> [3] Cho, Yae Jee, Jianyu Wang, and Gauri Joshi. "Client selection in federated learning: Convergence analysis and power-of-choice selection strategies." AISTATS (2022).
>
> [4] Gorbunov, Eduard, Filip Hanzely, and Peter Richtárik. "Local sgd: Unified theory and new efficient methods." International Conference on Artificial Intelligence and Statistics. PMLR, 2021.
>
> [5] Karimireddy, Sai Praneeth, et al. "Mime: Mimicking centralized stochastic algorithms in federated learning." arXiv preprint arXiv:2008.03606 (2020).
>
> [6] Karimireddy, Sai Praneeth, et al. "Scaffold: Stochastic controlled averaging for federated learning." International Conference on Machine Learning. PMLR, 2020.
>
> **2, Confusion about $\Gamma$ and $\Omega$ and incorrect understanding of “convergence rate**
>
> **Confusion about $\Gamma$ and $\Omega$:** $\Gamma$ and $\Omega$ are *independent* of time in Theorem 1 and the following Sec.3.3 because of the max and min over time operator as suggested in Theorem 1.  Sec.3.3 does not violate this assumption because what changes with time is the argument, $\Omega_i^t$, inside the max and min operator. For example, we defined $\Omega$  (Theorem 1) as
>
> $\Omega= \min_t \sum_{i=1}^N \rho_i^{t+1} (F_i(\bar{\mathbf{W}}^{t}) -F_i(\mathbf{W}^*_i)) \forall t\geq0$
>
> And in Sec 3.3 we introduced this $\Omega_i^t $, which is the argument of the min operation and changes with time.
>
> $\Omega_i^t \triangleq F_i(\bar{\mathbf{W}}^{t}) -F_i(\mathbf{W}^*_i)$
>
> We will further revise the paper to explain this.
>
> The confusion likely comes from the fact that Theorem 1 provides a *conservative* error bound (sec 3.2 title), meaning the worst case bound. Therefore, depending on the actual time-varying aggregation pattern, the actual convergence rate can be better. Sec 3.3 discusses how we can minimize $\Omega_i^t$ by using a different aggregation strategy. Because of the min operator over time, when $ \sum_{i=1}^N \rho_i^{t+1}\Omega_i^t$ is minimized, so is $\Omega$. Essentially, as a simple example, we are trying to minimize a constant $c=\min_t f(t)$ by minimizing $f(t)$. Therefore, there is no conflict in our description of $\Gamma$ and $\Omega$ in Theorem 1 and Sec. 3.3.

---

> > ### Author Response · Authors · 2022-11-15
> > **Cont'd**
> >
> > **Convergence rate:** Our understanding of convergence rate is exactly the *same* as the reviewer’s and is *correct*.  However, we believe that there is a misunderstanding on the optimal value in Theorem 1. Specifically, we completely agree with the reviewer’s definition of convergence rate: “convergence rate characterizes the finite-time error of some convergence metric (gap w.r.t. the optimal value of the problem in Theorem 1) w.r.t. T in Theorem 1.”
> >
> > We believe that the reviewer may take $F(\mathbf{W}^*)$ as the optimal value, whereas the actual optimal value is, for a given aggregation pattern, $F(\mathbf{W}^*) $+ **non-vanishing error**, which changes with different aggregation patterns and is the *optimal* value achievable for that aggregation pattern. In other words, convergence rate is how fast the error decreases with T [1], so the vanishing term measures convergence rate. In fact, a recent work [2] on client sampling, which we cited in the paper, also discussed convergence rate by decomposing the bound into a vanishing term and a non-vanishing term.  Therefore, in Theorem 1, the optimal value is:
> >
> > Optimal value in Theorem 1: $F(\mathbf{W}^*) + \frac{L}{\mu}(\Gamma -\Omega)$
> >
> > The reason we leave the non-vanishing term on the right-hand side is to highlight the speed and error trade-off, following the prior work [2].
> >
> > [1] Wang, Jianyu, et al. "A field guide to federated optimization." arXiv preprint arXiv:2107.06917 (2021).
> >
> > [2] Cho, Yae Jee, Jianyu Wang, and Gauri Joshi. "Client selection in federated learning: Convergence analysis and power-of-choice selection strategies." AISTATS (2022).
> >
> >
> > The reviewer might question why the algorithm does not converge to $F(\mathbf{W}^*)$ , which is the optimal value defined by proportional aggregation and when the non-vanishing error is zero. This is precisely the main contribution of this paper.  We found that an algorithm can achieve a stronger convergence rate. However, this comes at the theoretical cost of a non-vanishing convergence error w.r.t to the optimal value defined by proportional aggregation. Through exhaustive experiments, we have shown that this trade-off of speed for error is preferable in practice.
> >
> > **3, No theoretical proof for the practical algorithm**
> >
> > To adapt the theory to the real world, we have to make some practical approximations, which are not always possible to give an exact bound. This motivates us to conduct exhaustive experiments to validate the approximation assumption and the effectiveness of the algorithm beyond theory on large real world datasets. Furthermore, it is common to have two versions of an FL algorithm, a theoretical version and a practical version. This is true especially because theoretical analysis with **weighted averaging** is challenging and many works adopt uniform averaging in their derivation. For example, SCAFFOLD[1] and FedAdam[2] both have a theoretical version and a practical version with weighted averaging. We refer the reviewer to Appendix B in [2] for a more detailed discussion. The practical aggregation algorithm has been carefully validated on three types of heterogeneity to demonstrate its generality and compatibility with existing FL algorithms.
> >
> > [1] Karimireddy, Sai Praneeth, et al. "Scaffold: Stochastic controlled averaging for federated learning." International Conference on Machine Learning. PMLR, 2020
> >
> > [2] Reddi, Sashank, et al. "Adaptive federated optimization."ICLR (2021).

---

> > > ### Author Response · Authors · 2022-11-25
> > > **Thank you for your reviews!**
> > >
> > > Dear Reviewer cggb,
> > >
> > > We would like to provide a short summary of our response to your questions to further help you read the rebuttal.
> > >
> > > * We discussed reasons **why relaxing the strong convexity and bounded gradient assumptions is difficult** (Q1).
> > > * We clarified that $\Gamma$ and $\Omega$ are **time-independent**, and **our description is consistent** in the paper (Q2).
> > > * We clarified that **our understanding of the convergence rate is correct** and is the same as the reviewer’s (Q2).
> > > * We explained the **technical difficulty of providing theoretical analysis for the practical algorithm** (Q3).
> > >
> > > We hope these responses have answered your major concerns and happy holidays!
> > >
> > > Authors of 3299

---

### Official Review · Reviewer_cWRm · 2022-11-04

**Confidence:** 1
**Correctness:** 3
**Technical Novelty And Significance:** 3
**Empirical Novelty And Significance:** 3
**Recommendation:** 6

**Clarity, Quality, Novelty And Reproducibility:**

I checked the mathematical arguments of the main body and confirmed that there was no serious error. I could not check the detailed proof. There are several notations that are not defined well, e.g.,
- Definition 1 is strange because there is no clear definition of some notations, such as $F$. If this is not a standard in this literature, there should be clear definitions of such notations. Besides, the spaces of $\rho_i$ and $W$ are needed.
- On page 2: what is $\rho^t_i$. We can guess, but there is no definition.
- In Assumption 1, what is the definition of L2 norm? I firstly though that the expectation in the norm is taken over $W$ and $V$, but it is written that the assumption holds for all $W$ and $V$. I could not understand the definition of the norm.

**Strength And Weaknesses:**

The authors consider an intriguing problem in federated learning. Based on a finding on the proportional aggregation, the authors propose insightful framework, analysis, and methods. Unfortunately, I have not studied federated learning and cannot evaluate the significance. However, it seems that the theoretical results are sound, and the authors develop non-trivial methods based the theoretical results. In particular, the idea of time-varying aggregation is outstanding. Overall, I felt that the paper is high-quality and well-written.

**Summary Of The Paper:**

This paper considers federated learning with non-iid data. Under the non-iid data, standard methods do not work well. For this problem, the authors first consider the effectiveness of proportional aggregation. Then, they find a trade-off between convergence rate and convergence error. Based on this finding, they propose a novel method, called Exp-$\alpha$, which achieves the stronger convergence rates at the theoretical costs of a non-vanishing convergence error. Through several experiments, the authors confirm the empirical effectiveness.

**Summary Of The Review:**

Unfortunately, I am not familiar to this topic and cannot evaluate the significance of this paper. However, the paper is well written, and there is no serious error in mathematical statement .

---

> ### Author Response · Authors · 2022-11-15
> **Clarified confusion on notations**
>
> We thank the reviewer for the positive comments on the novelty of the paper. We would like to clarify some of the confusions in our notations.
>
> **1, F in definition 1**
>
> The F in definition refers to a generic objective function and is a standard notation in FL literature. For example, in standard classification tasks, $F(\mathbf{W})$ is instantiated as cross-entropy loss. We have made changes to help clarify this in the paper.
>
> **2, $\rho_i^t$ on page 2**
>
> Thank you for bringing this up. We have updated the paper to clarify this.  $\rho_i^t$ on page 2 refers to the time-varying aggregation weights. Specifically, $\rho_i^t$ is the aggregation weight of client $i$ at time $t$.
>
> **3, Definition of L2 norm**
>
> We follow the standard definition of L2 norm, i.e., square root of sum of squared values , in the definition of $\mu$-strongly convex function (Assumption 1).
>
> We appreciate the comments and the positive feedback.

---

### Author Response · Authors · 2022-11-15
**Summary of Rebuttal**

We appreciate all reviewers’ positive comments on the novelty and performance of the proposed method and comprehensiveness of experiments. In our rebuttals, we have:

- **[Experiments Added]** Added new experiments to discuss the effects of $\alpha$ w.r.t increasing heterogeneity and local update steps in appendix A5 (Ckau)
- **[Theory Added]** Added an extension of Theorem 1 to partial participation in appendix A12 (gbv4)
- **[Theory Added]**  Added theoretical proof that the convergence bound with proportional aggregation can be recovered from our more general bound in appendix A13 (gbv4).
- Clarified some of our notation (cWRm)
- Discussed technical questions on time-dependence and the definition of convergence rate (cggb)

We believe that these points address the concerns that the reviewers had, and we look forward to the discussion.

---

### Author Response · Authors · 2022-11-17
**A Gentle Reminder**

Dear Reviewers,

Thank you for your efforts in reviewing this paper. This is a gentle reminder that we are almost approaching the end of the discussion period. We have provided clarifications and additional experiments/theory to answer your questions. Please let us know if you have any other questions.

Best,
Authors of Paper 3299.

---

### Decision · Program_Chairs · 2023-01-20

**Decision:**

Reject

**Justification For Why Not Higher Score:**

see above

**Justification For Why Not Lower Score:**

see above

**Metareview: Summary, Strengths And Weaknesses:**

There was an extensive discussion with the reviewers about this paper but all reviewers believed that the paper was not ready for publication in its current form.  The core issues were summarized by a reviewer Cggb, in particular comment 3. We discussed for example the authors rebuttal of this point on Nov 14th, but it was unanimously agreed that this was not sufficient. In general, a good theory is meant to be something that applies when an algorithm is applied, but if it is the case that the approximations, or heuristics, used to get the algorithm to work in practice render the theory inapplicable, then it puts into question the purpose of the theory. We also felt, especially, since several of the reviewers all had trouble understanding $\Gamma$ and $\Omega$ that in the original submission more details should have been placed into ensuring that there was no chance of confusion regarding these terms and how they apply. As stated, for example, with $\Omega$ is a minimization over all time and indeed while this is independent of time, it makes it not possible to evaluate in practice, and it would have been useful to have more discussion on the additional assumptions needed for this to work in practice. All the reviewers felt that with some work, the paper would likely be an accept in the next conference. We encourage you to revise the paper, clearly incorporate the additional information you provided in your rebuttal into the next version to help to avoid any confusions, and re-submit to the next conference.

**Summary Of Ac-Reviewer Meeting:**

see above